# Visual Search Asymmetry:
# Deep Nets and Humans Share Similar Inherent Biases

Shashi Kant Gupta[1], Mengmi Zhang[2,3], Chia-Chien Wu[4], Jeremy M. Wolfe[4], and Gabriel Kreiman[2,3]

[1]Indian Institute of Technology Kanpur, India
[2]Children's Hospital, Harvard Medical School
[3]Center for Brains, Minds and Machines
[4]Brigham and Women's Hospital, Harvard Medical School
Address correspondence to gabriel.kreiman@tch.harvard.edu

## Abstract

Visual search is a ubiquitous and often challenging daily task, exemplified by looking for the car keys at home or a friend in a crowd. An intriguing property of some classical search tasks is an asymmetry such that finding a target A among distractors B can be easier than finding B among A. To elucidate the mechanisms responsible for asymmetry in visual search, we propose a computational model that takes a target and a search image as inputs and produces a sequence of eye movements until the target is found. The model integrates eccentricity-dependent visual recognition with target-dependent top-down cues. We compared the model against human behavior in six paradigmatic search tasks that show asymmetry in humans. Without prior exposure to the stimuli or task-specific training, the model provides a plausible mechanism for search asymmetry. We hypothesized that the polarity of search asymmetry arises from experience with the natural environment. We tested this hypothesis by training the model on augmented versions of ImageNet where the biases of natural images were either removed or reversed. The polarity of search asymmetry disappeared or was altered depending on the training protocol. This study highlights how classical perceptual properties can emerge in neural network models, without the need for task-specific training, but rather as a consequence of the statistical properties of the developmental diet fed to the model. All source code and data are publicly available at `https://github.com/kreimanlab/VisualSearchAsymmetry`.

## 1 Introduction

Humans and other primates continuously move their eyes in search of objects, food, or friends. Psychophysical studies have documented how visual search depends on the complex interplay between the target objects, search images, and the subjects' memory and attention [49, 29, 37, 44, 4, 18, 30]. There has also been progress in describing the neurophysiological steps involved in visual processing [32, 39, 23] and the neural circuits that orchestrate attention and eye movements [14, 38, 31, 10, 3].

A paradigmatic and intriguing effect is *visual search asymmetry*: searching for an object, A, amidst other objects, B, can be substantially easier than searching for object B amongst instances of A. For example, detecting a curved line among straight lines is faster than searching for a straight line among curved lines. Search asymmetry has been observed in a wide range of tasks [48, 22, 46, 47, 34, 50, 43]. Despite extensive phenomenological characterization [34, 43, 15, 8], the mechanisms underlying how neural representations guide visual search and lead to search asymmetry remain mysterious.

35th Conference on Neural Information Processing Systems (NeurIPS 2021).

Inspired by prior works on visual search modelling [7, 21, 19, 51, 25], we developed an image-computable model (eccNET) to shed light on the fundamental inductive biases inherent to neural computations during visual search. The proposed model combines eccentricity-dependent sampling, and top-down modulation through target-dependent attention. In contrast to deep convolutional neural networks (CNNs) that assume uniform resolution over all locations, we introduce eccentricity-dependent pooling layers in the visual recognition processor, mimicking the separation between fovea and periphery in the primate visual system. As the task-dependent target information is essential during visual search [14, 3, 49, 51], the model stores the target features and uses them in a top-down fashion to modulate unit activations in the search image, generating a sequence of eye movements until the target is found.

We examined six foundational psychophysics experiments showing visual search asymmetry and tested eccNET on the same stimuli. Importantly, the model was pre-trained for object classification on ImageNet and was *not* trained with the target or search images, or with human visual search data. The model spontaneously revealed visual search asymmetry and qualitatively matched the polarity of human behavior. We tested whether asymmetry arises from the natural statistics seen during object classification tasks. When eccNET was trained from scratch on an augmented ImageNet with altered stimulus statistics. e.g., rotating the images by 90 degrees, the polarity of search asymmetry disappeared or was modified. This demonstrates that, in addition to the model's architecture, inductive biases in the developmental training set also govern complex visual behaviors. These observations are consistent with, and build bridges between, psychophysics observations in visual search studies [35, 40, 7, 2, 8] and analyses of behavioral biases of deep networks [28, 53, 33].

## 2   Psychophysics Experiments in Visual Search Asymmetry

We studied six visual search asymmetry psychophysics experiments [48, 22, 46, 47] (**Figure 1**). Subjects searched for a target intermixed with distractors in a search array (see **Appendix A** for stimulus details). There were target-present and target-absent trials; subjects had to press one of two keys to indicate whether the target was present or not.

**Experiment 1 (Curvature)** involved two conditions (**Figure 1A**) [48]: searching for (a) a straight line among curved lines, and (b) a curved line among straight lines. The target and distractors were presented in any of the four orientations: -45, 0, 45, and 90 degrees.

**Experiment 2 (Lighting Direction)** involved two conditions (**Figure 1B**) [22]: searching for (a) left-right luminance changes among right-left luminance changes, and (b) top-down luminance changes among down-top luminance changes. There were 17 intensity levels.

**Experiments 3-4 (Intersection)** involved four conditions (**Figure 1C-D**) [46]: searching for (a) a cross among non-crosses, (b) a non-cross among crosses, (c) a rotated L among rotated Ts, and (d) a rotated T among rorated Ls. The objects were presented in any of the four orientations: 0, 90, 180, and 270 degrees.

**Experiments 5-6 (Orientation)** involved four conditions (**Figure 1E-F**) [47]: searching for (a) a vertical line among 20-degrees-tilted lines, (b) a 20-degree-tilted line among vertical straight lines, (c) a 20-degree tilted line among tilted lines from -80 to 80 degrees, and (d) a vertical straight line among tilted lines from -80 to 80 degrees.

## 3   Eccentricity-dependent network (eccNET) model

A schematic of the proposed visual search model is shown in **Figure 2**. eccNET takes two inputs: a target image ($I_t$, object to search) and a search image ($I_s$, where the target object is embedded amidst distractors). eccNET starts fixating on the center of $I_s$ and produces a sequence of fixations. eccNET uses a pre-trained 2D-CNN as a proxy for the ventral visual cortex, to extract eccentricity-dependent visual features from $I_t$ and $I_s$. At each fixation $n$, these features are used to calculate a top-down attention map ($A_n$). A winner-take-all mechanism selects the maximum of the attention map $A_n$ as the location for the $n + 1$-th fixation. This process iterates until eccNET finds the target with a total of $N$ fixations. eccNET has infinite inhibition of return and therefore does not revisit previous locations. Humans do *not* have perfect memory and do re-visit previously fixated locations [52]. However, in the 6 experiments considered here $N$ is small; therefore, the probability of return fixations is small

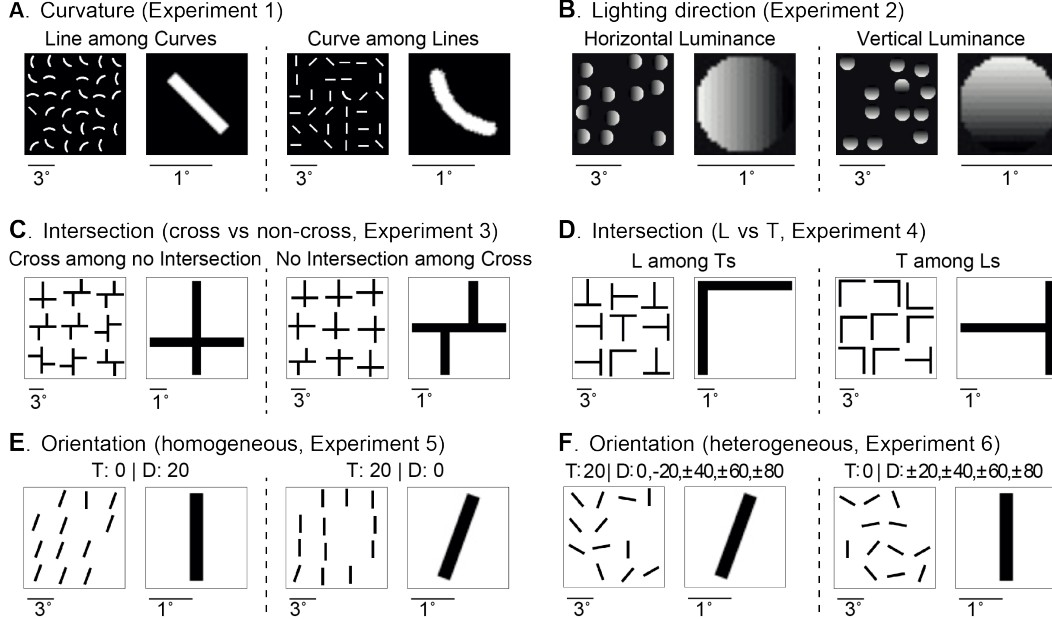

Figure 1: **Schematic illustration of the 6 experiments for asymmetry in visual search.** Each experiment has two conditions; in each condition, an example search image (left) and target image (right) are shown. The target image is smaller than the search image (see scale bars below each image). **A. Experiment 1.** Searching for a curve among lines and vice versa ([48]). **B. Experiment 2.** Searching for vertical luminance changes among horizontal luminance changes and vice versa ([22]). **C. Experiment 3.** Searching for shapes with no intersection among crosses and vice versa ([46]). **D. Experiment 4.** Searching for rotated Ts among rotated Ls and vice versa ([46]). **E. Experiment 5.** Searching for oblique lines with fixed angles among vertical lines and vice versa ([47]). **F. Experiment 6.** Similar to Experiment 5 but using oblique lines of different orientations ([47]). In all cases, subjects find the target faster in the condition on the right.

and the effect of limited working memory capacity is negligible. eccNET needs to verify whether the target is present at each fixation. Since we focus here on visual search (target localization instead of recognition), we simplify the problem by bypassing the target verification step and using an "oracle" recognition system [26]. The oracle checks whether the selected fixation falls within the ground truth target location, defined as the bounding box of the target object. We only consider target-present trials for the model and therefore eccNET will always find the target.

Compared with the invariant visual search network (IVSN, [51]), we highlight novel components in eccNET, which we show to be critical in Section 4:

1. Standard 2D-CNNs, such as VGG16 [41], have uniform receptive field sizes (pooling window sizes) within each layer. In stark contrast, visual cortex shows strong eccentricity-dependent receptive field sizes. Here we introduced eccentricity-dependent pooling layers, replacing all the max-pooling layers in VGG16.

2. Visual search models compute an attention map to decide where to fixate next. The attention map in IVSN did not change from one fixation to the next. Because the visual cortex module in eccNET changes with the fixation location in an eccentricity-dependent manner, here the attentional maps ($A_n$) are updated at each fixation $n$.

3. In contrast to IVSN where the top-down modulation happens only in a single layer, eccNET combines top-down modulated features across multiple layers.

**Eccentricity-dependent pooling in visual cortex in eccNET**. Receptive field sizes in visual cortex increase from one brain area to the next (**Figure 3B**, right). This increase is captured by current visual recognition models through pooling operations. In addition, receptive field sizes also increase with eccentricity *within* a given visual area [16] (**Figure 3B**, right). Current 2D-CNNs assume uniform sampling within a layer and do not reflect this eccentricity dependence. In contrast, we introduced eccentricity-dependent pooling layers in eccNET. Several psychophysics observations [8, 35] and

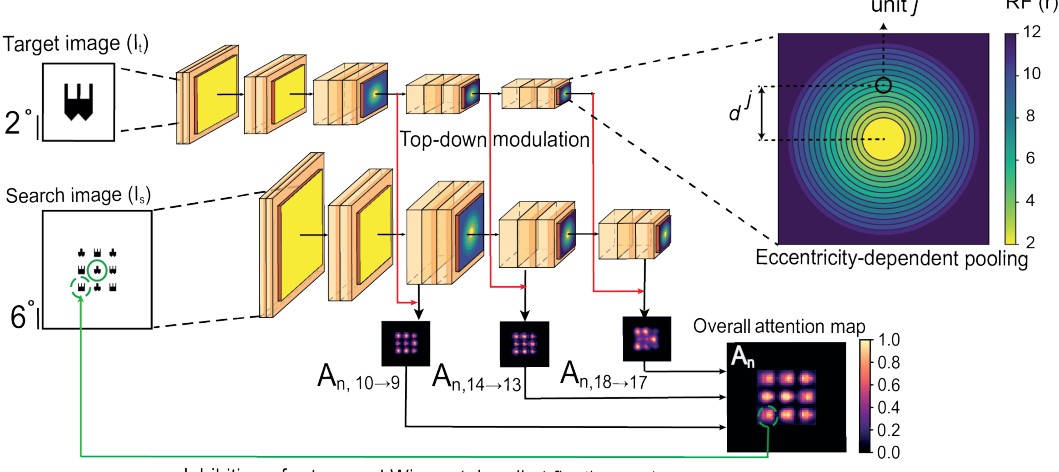

Figure 2: **Schematic of the computational model of visual search.** The model takes as input a target image ($I_t$) and a search image ($I_s$), both of which are processed through the same 2D-CNN with shared weights. At each fixation $n$, the model produces a top-down modulation map ($A_n$) that directs the next eye movement (Section 3). The color bar on the right denotes attention values. Instead of using a typical 2D-CNN with uniform pooling window sizes, here we introduce an eccentricity-dependent sampling within each pooling layer of the network by modifying the VGG16 ([41]) architecture. The upper right inset illustrates eccentricity-dependent pooling layer $l$. It shows the receptive field sizes ($r^j_{l,n}$) for each unit $j$ with distance $d^j$ from the centre (the color bar denotes the size of the pooling window in pixels). See **Figure S19** for example eye movement patterns from eccNet in each experiment.

works on neuro-anatomical connectivity [42, 45, 11, 12, 6] are consistent with the enhancement of search asymmetry by virtue of eccentricity-dependent sampling. We first define notations used in standard average pooling layers [17]. For simplicity, we describe the model components using pixels (**Appendix D** shows scaling factors used to convert pixels to degrees of visual angles (dva) to compare with human behavior in Section 2). The model does not aim for a perfect quantitative match with the macaque neurophysiological data, but rather the goal is to preserve the trend of eccentricity versus receptive field sizes (see **Appendix J** for further discussion).

Traditionally, unit $j$ in layer $l+1$ of VGG16 takes the average of all input units $i$ in the previous layer $l$ within its local receptive field of size $r_{l+1}$ and its activation value $y$ is given by: $y^j_{l+1} = \frac{1}{r_{l+1}} \sum_{i=0}^{r_{l+1}} y^i_l$ In the eccentricity-dependent operation, the receptive field size $r^j_{l+1,n}$ of input unit $j$ in layer $l + 1$ is a linear function of the Euclidean distance $d^j_{l+1,n}$ between input unit $j$ and the current fixation location ($n$) on layer $l + 1$.

$$r^j_{l+1,n} = \begin{cases} \lfloor \eta_{l+1} \gamma_{l+1}(d^j_{l+1,n}/\eta_{l+1} - \delta) + 2.5 \rfloor, & \text{if } d^j_{l+1,n}/\eta_{l+1} > \delta \\ 2, & \text{if } d^j_{l+1,n}/\eta_{l+1} < \delta \end{cases} \tag{1}$$

The floor function $\lfloor \cdot \rfloor$ rounds down the window sizes. The positive scaling factor $\gamma_{l+1}$ for layer $l + 1$ defines how fast the receptive field size of unit $j$ expands with respect to its distance from the fixation at layer $l + 1$. The further away the unit $j$ is from the current fixation, the larger the receptive field size (**Figure 2B**; in this figure, the fixation location is at the image center). Therefore, the resolution is highest in the fixation location and decreases in peripheral regions. Based on the slope of eccentricity versus receptive field size in the macaque visual cortex [16], we experimentally set $\gamma_3 = 0.00$, $\gamma_6 = 0.00$, $\gamma_{10} = 0.14$, $\gamma_{14} = 0.32$, and $\gamma_{18} = 0.64$. See **Figure 3B** for the slopes of eccentricity versus receptive field sizes over pooling layers. We define $\delta$ as the constant fovea size. For those units within the fovea, we set a constant receptive field size of 2 pixels. Constant $\eta_{l+1}$ is a positive scaling factor which converts the dva of the input image to the pixel units at the layer $l$ (see **Appendix D** for specific values of $\eta_{l+1}$). As in the stride size in the original pooling layers of VGG16, we empirically set a constant stride of 2 pixels for all eccentricity-dependent pooling layers.

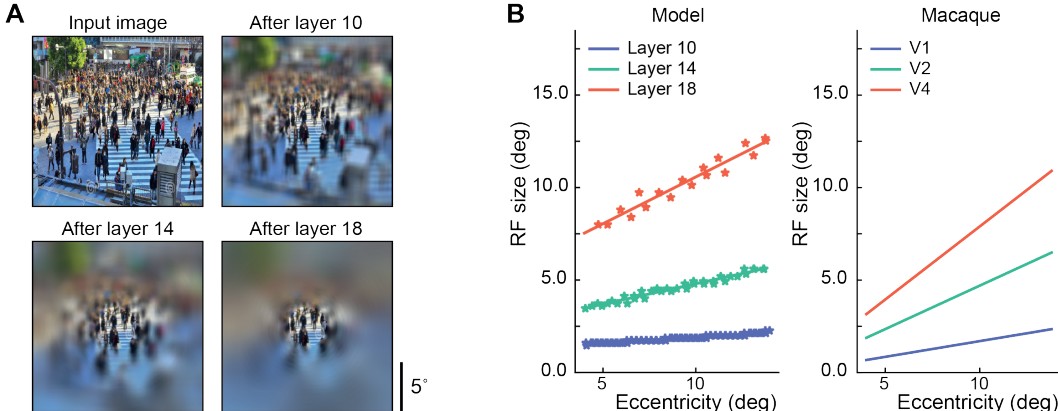

Figure 3: **Eccentricity dependence in the model matches that in macaque monkey visual cortex.**
**A.** Visualization of one example image and its corresponding eccentricity-dependent sampling at layers 10, 14, and 18 (see **Appendix E** for implementation). The figures are best viewed after zooming in to assess the small amount of blurring in the center. **B.** Eccentricity-dependent sampling leads to increasing receptive field sizes as a function of eccentricity, in addition to increased receptive field sizes across layers for the model (left) and also for the macaque visual cortex ([16], see **Appendix F** for implementation).

A visualization for $r_{l,n}^j$ is shown in **Figure 2**, where different colors denote how the receptive field sizes expand within a pooling layer. Receptive field sizes $r_{l,n}^j$ versus eccentricity $d_{l,n}^j$ in pixels at each pooling layer are reported in **Table S1**. **Figure 3A** illustrates the change in acuity at different pooling layers. These customized eccentricity-dependent pooling layers can be easily integrated into other object recognition deep neural networks. All the computational steps are differentiable and can be trained end-to-end with other layers. However, since our goal is to test the generalization of eccNET from object recognition to visual search, we did not retrain eccNET and instead, we used the pre-trained weights of VGG16 on the ImageNet classification task.

**Top-down attention modulation across multiple layers of visual cortex in eccNET.** Previous works have described top-down modulation in simplified models for search asymmetry [21, 7]. Here we focus on developing a top-down modulation mechanism in deep networks. Given the current $n^{th}$ fixation location, the visual cortex of eccNET with eccentricity-dependent pooling layers extracts feature maps $\phi_{l+1}^t$ at layer $l+1$ for the target image $I_t$. Correspondingly, the model extracts $\phi_{l,n}^s$ in response to the search image $I_s$. Inspired by the neural circuitry of visual search [9, 51], we define the top-down modulation map $A_{l+1\rightarrow l,n}$ as:

$$A_{l+1\rightarrow l,n} = m(\phi_{l+1}^t, \phi_{l,n}^s) \qquad (2)$$

where $m(\cdot)$ is the target modulation function defined as a 2D convolution with $\phi_{l+1}^t$ as convolution kernel operating on $\phi_{l,n}^s$ and $\phi_{l+1}^t$ is the feature map after pooling operation on $\phi_l^t$. Note that the layer $l+1$ modulates the activity of layer $l$. Following the layer conventions in TensorFlow Keras [1], we empirically selected $l = 9, 13, 17$ as the layers where top-down modulation is performed (see **Figure S23** for exact layer numbering).

To compute the overall top-down modulation map $A_n$, we first resize $A_{10\rightarrow 9,n}$ and $A_{14\rightarrow 13,n}$ to be of the same size as $A_{18\rightarrow 17,n}$. eccNET then takes the weighted linear combination of normalized top-down modulation maps across all three layers: $A_n = \sum_{l=9,13,17} w_{l,n} \frac{(A_{l+1\rightarrow l,n} - \min A_{l+1\rightarrow l,n})}{(\max A_{l+1\rightarrow l,n} - \min A_{l+1\rightarrow l,n})}$

where $w_{l,n}$ are weight factors governing how strong top-down modulation at the $l$th layer contributes to the overall attention map $A_n$. Each of the top-down attention maps contributes different sets of unique features. Since these features are specific to the target image type, the weights are not necessarily equal and they depend on the demands of the given task. One way to obtain these weights would be by parameter fitting. Instead, to avoid parameter fitting specific to the given visual search experiment, these weights $w_{l,n}$ were calculated during the individual search trials using the maximum activation value from each individual top-down modulation map: $w_{l,n} = (\max A_{l+1\rightarrow l,n})/(\sum_{i=9,13,17} \max A_{i+1\rightarrow i,n})$. A higher activation value will mean a high similarity

between the target and search image at the corresponding feature level. Thus, a higher weight to the attention map produced using that feature layer ensures that the model assigns higher importance to the attention maps at those feature layers that are more prominent in the target image (see **Figure S22** and **Appendix I** for examples on how the attention map is computed).

**Comparison with human performance and baseline models.** The psychophysics experiments in Section 2 did not measure eye movements and instead report a *key press reaction time* (RT) in milliseconds when subjects detected the target. To compare fixation sequences predicted by eccNET with reaction times measured in human experiments, we conducted a similar experiment as Experiment 4 in Section 2 using eye tracking (**Figure S1**, **Appendix C**). Since RT results from a combination of time taken by eye movements plus a finger motor response time, we performed a linear least-squares regression (**Appendix B**) on the eye tracking and RT data collected from this experiment. We fit $RT$ as a function of number of fixations $N$ until the target was found: $RT = \alpha * N + \beta$. The slope $\alpha = 252.36$ ms/fixation approximates the duration of a single saccade and subsequent fixation, and the intercept $\beta = 376.27$ can be interpreted as a finger motor response time. We assumed that $\alpha$ and $\beta$ are approximately independent of the task. We used the same fixed values of $\alpha$ and $\beta$ to convert the number of fixations predicted by eccNET to RT throughout all 6 experiments.

We introduced two evaluation metrics. First, we evaluated the model and human performance showing **key press reaction times (RT)** as a function of number of items on the stimulus, as commonly used in psychophysics [48, 22, 46, 47] (Section 4). We compute the slope of the RT versus number of items plots for the hard ($H$, larger search slopes) and easy ($E$, lower search slopes) conditions within each experiment. We define the **Asymmetry Index**, as $(H - E)/(H + E)$ for each experiment. If a model follows the human asymmetry patterns for a given experiment, it will have a positive Asymmetry Index. The Asymmetry Index takes a value of 0 if there is no asymmetry, and a negative value indicates that the model shows the opposite behavior to humans.

We included four baselines for comparison with eccNET:

**Chance:** a sequence of fixations is generated by uniform random sampling.

**pixelMatching:** the attention map is generated by sliding the raw pixels of $I_t$ over $I_s$ (stride $= 1 \times 1$).

**GBVS [19]:** we used bottom-up saliency as the attention map.

**IVSN [51]:** the top-down attention map is based on the features from the top layer of VGG16.

## 4 Results

**eccNET predicts human fixations in visual search**: The model produces a sequence of eye movements in every trial (see **Figure S19** for example eye movement patterns by eccNet in each experiment). The original asymmetry search experiments (**Figure1**) did not measure eye movements. Thus we repeated the L vs. T search experiment (Experiment 4) to measure eye movements. We also used data from three other search tasks to compare the fixation patterns [51]. We compared the fixation patterns between humans and EccNet in terms of three metrics [51]: 1. number of fixations required to find the target in each trial (the cumulative probability distribution $p(n)$ that the subject or model finds the target within $n$ fixations); 2. scanpath similarity score, which compares the spatiotemporal similarity in fixation sequences [5, 51]; and 3. the distribution of saccade sizes. The results show that the model approximates the fixations made by humans on a trial-by-trial basis both in terms of the number of fixations and scanpath similarity. The model also presents higher consistency with humans in the saccade distributions than any of the previous models in [51] (**Figures S15-18**).

**eccNET qualitatively captures visual search asymmetry**. **Figure 4A** (left, red line) shows the results of Experiment 1 (**Figure 1A**, left), where subjects looked for a straight line amongst curved lines. Increasing the number of distractors led to longer reaction times (RTs), as commonly observed in classical visual search studies. When the target and distractors were reversed and subjects had to search for a curved line in the midst of straight lines, the RTs were lower and showed minimal dependence on the number of objects (**Figure 4A**, left, blue line). In other words, it is easier to search for a curved line amidst straight lines than the reverse.

The same target and search images were presented to eccNET. Example fixation sequences from the model for Experiment 1 are shown in **Figure S19A**. **Figure 4A** (right) shows eccNET's RT as a function of the number of objects for Experiment 1. eccNET had not seen any of these types of

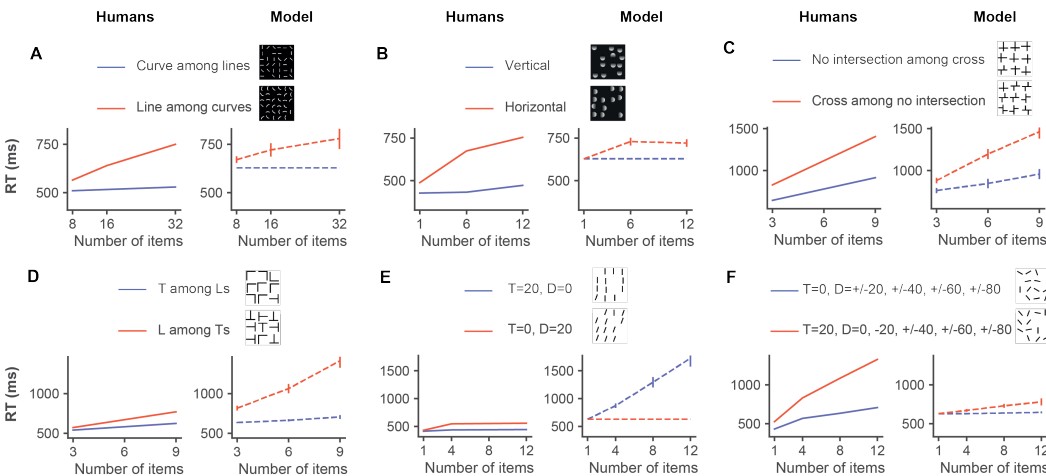

Figure 4: **The model shows visual search asymmetry and qualitatively captures human behavior.** Reaction time as a function of the number of items in the display for each of the six experiments in **Figure 1** for humans (left, solid lines) and for the model (right, dashed lines). The results for humans were obtained from [48, 22, 46, 47]. The line colors denote the different experimental conditions. Error bars for the model denote standard error (SE). Error bars were not available in the original publications for the human data.

images before and the model was never trained to search for curves or straight lines. Critically, there was no explicit training to show an asymmetry between the two experimental conditions. Remarkably, eccNET qualitatively captured the key observations from the psychophysics experiment [48]. When searching for curves among straight lines (blue), the RTs were largely independent of the number of distractors. In contrast, when searching for straight lines among curves (red), the RTs increased substantially with the number of distractors. The model's absolute RTs were not identical to human RTs (more discussions in Section 5). Of note, there was no training to quantitatively fit the human data. In sum, without any explicit training, eccNET qualitatively captures this fundamental asymmetry in visual search behavior.

We considered five additional experiments (Section 2) that reveal similar asymmetries on a wide range of distinct features (**Figure 1B-F**). In Experiment 2, it is easier to search for top-down luminance changes than left-right luminance changes (**Figure 1B**; [22]). In Experiments 3-4, it is easier to find non-intersections among crosses than the reverse (**Figure 1C**; [46]), and it is easier to find a a rotated letter T among rotated letters L than the reverse (**Figure 1D**; [46]). In Experiments 5-6, it is easier to find a tilted bar amongst vertical distractors than the reverse when distractors are homogeneous (**Figure 1E**; [47]) but the situation reverses when the distractors are heterogenous (**Figure 1E-F**; [47]). In all of these cases, the psychophysics results reveal lower RTs and only a weak increase in RTs with increasing numbers of distractors for the easier search condition and a more pronounced increase in RT with more distractors for the harder condition (**Figure 4A-F**, left panels).

Without any image-specific or task-specific training, eccNET qualitatively captured these asymmetries (**Figure 4A-D, F**, right panels). As noted in Experiment 1, eccNET did not necessarily match the human RTs at a quantitative level. While in some cases the RTs for eccNET were comparable to humans (e.g., **Figure 4C**), in other cases, there were differences (e.g., **Figure 4D**). Despite the shifts along the y-axis in terms of the absolute RTs, eccNET qualitatively reproduced the visual search asymmetries in 5 out of 6 experiments. However, in Experiment 5 where humans showed a minimal search asymmetry effect, eccNET showed the opposite behavior (more discussions in Section 5).

The slope of RT versus number of object plots is commonly used to evaluate human search efficiency. We computed the Asymmetry Index (Section 3) to compare the human and model results. Positive values for the asymmetry index for eccNET indicate that the model matches human behavior. There was general agreement in the Asymmetry Indices between eccNET and humans, except for Experiment 5 (**Figure 5A**). Since the Asymmetry Index is calculated using the search slopes (Section 3), it is independent of shifts along the y-axis in **Figure 4**. Thus, the eccNET Asymmetry Indices can match the human indices regardless of the agreement in the absolute RT values.

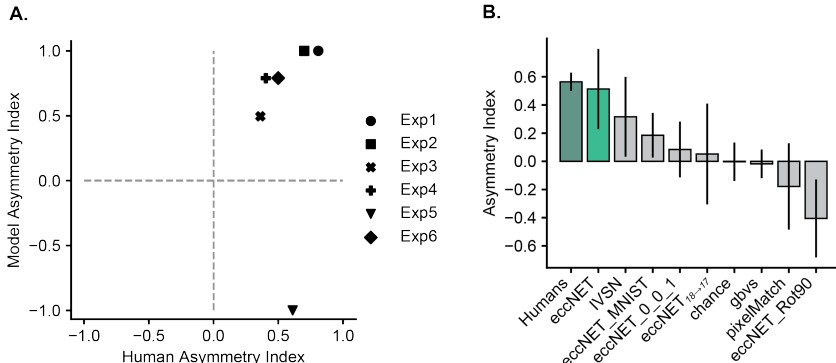

Figure 5: **EccNet outperforms other baselines and ablation demonstrates the relevance of its components. A.** Asymmetry Index (Section 3) for eccNet (y-axis) versus humans (x-axis) for each experiment. Points in the 1st quadrant indicate that eccNet follows the human asymmetry patterns. See **Figure S14** for Asymmetry Index plots of other models. **B.** Comparison to ablated and alternative models. Average Asymmetry Index for humans (dark green), eccNET (light green), and alternative models (light gray) defined in Section 3. The error bars show the standard error (SE) over all 6 experiments.

**Baseline models fail to capture search asymmetry**: We compared the performance of eccNET with several baseline models (**Figure 5B**, see also **Figures S2-S5** for the corresponding RT versus number of objects plots). All the baseline models also had infinite inhibition of return, oracle recognition, and the same method to convert number of fixations into RT values.

As a null hypothesis, we considered a model where fixations landed on random locations. This model consistently required longer reaction times and yielded an Asymmetry Index value close to zero, showing no correlation with human behavior (**Figure 5B**, **Figure S3**, *chance*). A simple template-matching algorithm also failed to capture human behavior (**Figure 5B**, **Figure S5** *pixelMatch*), suggesting that mere pixel-level comparisons are insufficient to explain human visual search. Next, we considered a purely bottom-up algorithm that relied exclusively on saliency (**Figure 5B**, **Figure S4**, *GBVS*); the failure of this model shows that it is important to take into account the top-down target features to compute the overall attention map.

None of these baseline models contain complex features from natural images. We reasoned that the previous IVSN architecture [51], which was exposed to natural images through ImageNet, would yield better performance. Indeed, IVSN showed a higher Asymmetry Index than the other baselines, yet its performance was below that of eccNET (**Figure 5B**, **Figure S2**, *IVSN*). Thus, the components introduced in eccNET play a critical role; next, we investigated each of these components.

**Model ablations reveal essential components contributing to search asymmetry**: We systematically ablated two essential components of eccNET (**Figure 5B**). In eccNET, top-down modulation occurs at three levels: layer 10 to layer 9 ($A_{10 \to 9}$), 14 to 13 ($A_{14 \to 13}$), and 18 to 17 ($A_{18 \to 17}$). We considered eccNET$_{18 \to 17}$, where top-down modulation only happened at layer 18 to 17 ($A_{18 \to 17}$). eccNET$_{18 \to 17}$ yielded a lower average Asymmetry Index (0.084), suggesting that visual search benefits from the combination of layers for top-down attention modulation (see also **Figure S7**).

To model the distinction between foveal and peripheral vision, we introduced eccentricity-dependent pooling layers. The resulting increase in receptive field size is qualitatively consistent with the receptive field sizes of neurons in the macaque visual cortex (**Figure 3B**). To assess the impact of this step, we replaced all eccentricity-dependent pooling layers with the max-pooling layers of VGG16 (eccNET$_{noecc}$ ). The lower average Asymmetry Index implies that eccentricity-dependent pooling is an essential component in eccNET for asymmetry in visual search (see also **Figure S6**). Moreover, to evaluate whether asymmetry depends on the architecture of the recognition backbone, we tested ResNet152 [20] on the six experiments (**Figure S24**). Though the ResNet152 backbone does not approximate human behaviors as well as eccNet with VGG16, it still shows similar asymmetry behavior (positive asymmetry search index) in four out of the six experiments implying search asymmetry is a general effect for deep networks.

**The statistics of training data biases polarity of search asymmetry**. In addition to the ablation experiments, an image-computable model enables us to further dissect the mechanisms responsible

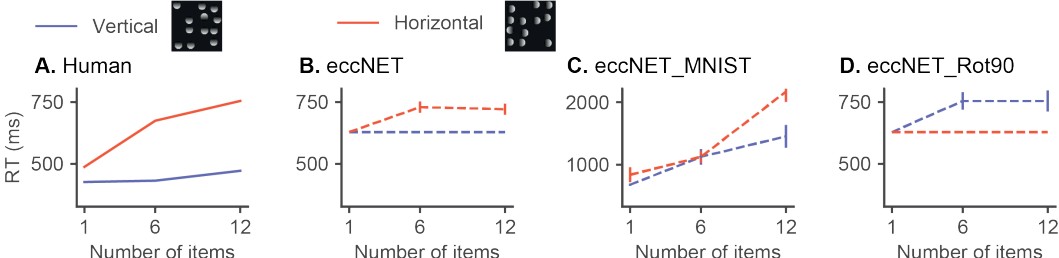

Figure 6: **Training data biases polarity of search asymmetry.** Reaction times as a function of the number of items for Experiment 2 on illumination conditions. (**A-B**) Humans and eccNET (same results as in **Figure 4B** reproduced here for comparison purposes). (**C**) eccNET trained on the MNIST [13]. (**D**) eccNET trained on ImageNet [24] after 90 degree rotation.

for visual search asymmetry by examining the effect of training. Given that the model was not designed, or explicitly trained, to achieve asymmetric behavior, we were intrigued by how asymmetry could emerge. We hypothesized that asymmetry can be acquired from the features learnt through the natural statistics of the training images. To test this hypothesis, as a proof-of-principle, we first focused on Experiment 2 (**Figure 1B**).

We trained the visual cortex of eccNET from scratch using a training set with completely different image statistics. Instead of using ImageNet containing millions of natural images, we trained eccNET on MNIST [13], which contains grayscale images of hand-written digits. We tested eccNET_MNIST in Experiment 2 (**Appendix G2**). The asymmetry effect of eccNET_MNIST disappeared and the absolute reaction times required to find the target increased substantially (**Figure 6C**). The average Asymmetry Index was also significantly reduced (**Figure 5B**, **Figure S8**).

To better understand which aspects of natural image statistics are critical for asymmetry to emerge, we focused on the distinction between different lighting directions. We conjectured that humans are more used to lights coming from a vertical than a horizontal direction and that this bias would be present in the ImageNet dataset. To test this idea, we trained eccNET from scratch using an altered version of ImageNet where the training images were rotated by 90 degrees counter-clockwise, and tested eccNET on the same stimuli in Experiment 2 (see **Appendix G1** for details). Note that random image rotation is not a data augmentation step for the VGG16 model pre-trained on ImageNet [41]; thus, the VGG16 pre-trained on ImageNet was only exposed to standard lighting conditions. Interestingly, eccNET trained on the 90-degree-rotated ImageNet (eccNET_Rot90) shows a polarity reversal in asymmetry (**Figure 6**, **Figure S9**), while the absolute RT difference between the two search conditions remains roughly the same as eccNET.

We further evaluated the role of the training regime in other experiments. First, we trained the model on Imagenet images after applying a "fisheye" transform to reduce the proportion of straight lines and increase the proportion of curves. Interestingly we observed a reversal in the polarity of asymmetry for Experiment 1 (curves among straight lines search) while the polarity for other experiments remained unchanged (**Figure S10**). Second, we introduced extra vertical and horizontal lines in the training data, thus increasing the proportion of straight lines. Since reducing the proportion of straight lines reverses the polarity for "Curve vs Lines", we expected that increasing the proportion of straight lines might increase the Asymmetry Index. However, the polarity for asymmetry did not change for Experiment 1 (**Figure S11**). Third, to test whether dataset statistics other than Imagenet would alter the polarity, we also trained the model on the Places 365 dataset and rotated Place 365 dataset. We found this manipulation altered some of the asymmetry polarities but not others, similar to the experiment done on MNIST dataset. Unlike the results using the MNIST case, the absolute reaction times required to find the target did not increase significantly (**Figures S12-13**). In sum, both the architecture and the training regime play an important role in visual search behavior. In most cases, but not in all manipulations, the features learnt during object recognition are useful for guiding visual search and the statistics from the training images contribute to visual search asymmetry.

## 5 Discussion

We examined six classical experiments demonstrating asymmetry in visual search, whereby humans find a target, A, amidst distractors, B, much faster than in the reverse search condition of B among A.

Given the similarity between the target and distractors, it is not clear why one condition should be easier than the other. Thus, these asymmetries reflect strong priors in how the visual system guides search behavior. We propose eccNET, a visual search model with eccentricity-dependent sampling and top-down attention modulation. At the heart of the model is a "ventral visual cortex" module pre-trained on ImageNet for classification [36]. The model had no previous exposure to the images in the current study, which are not in ImageNet. Moreover, the model *was not trained in any of these tasks*, did not have any tuning parameters dependent on eye movement or reaction time (RT) data, and was not designed to reveal asymmetry. Strikingly, despite this lack of tuning or training, asymmetric search behavior emerged in the model. Furthermore, eccNET captures observations in five out of six human psychophysics experiments in terms of the search costs and Asymmetry Index.

Image-computable models allow us to examine potential mechanisms underlying asymmetries in visual search. Even though the target and search images are different from those in ImageNet, natural images do contain edges of different size, color, and orientation. To the extent that the distribution of image statistics in ImageNet reflects the natural world, a question that has been contested and deserves further scrutiny, one might expect that the training set could capture *some* priors inherent to human perception. We considered this conjecture further, especially in the case of Experiment 2. The asymmetry in vertical versus horizontal illumination changes has been attributed to the fact that animals are used to seeing light coming from the top (the sun), a bias likely to be reflected in ImageNet. Consistent with this idea, changing the training diet for eccNET alters its visual search behavior. Specifically, rotating the images by 90 degrees, altering the illumination direction used to train eccNET, led to a reversal in the polarity of search asymmetry. Modifications in search asymmetry in other tasks were sometimes also evident upon introducing other changes in the training data such as modifying the proportion of straight lines, or using the places365 or MNIST datasets (**Figure 6**, **S8, S10-13**). However, the training regime is not the only factor that governs search asymmetry. For example, the network architecture also plays a critical role (**Figure 5B**).

Although eccNET *qualitatively* captures the critical observations in the psychophysics experiments, the model does not always yield accurate estimates of the absolute RTs. Several factors might contribute towards the discrepancy between the model and humans. First and foremost, eccNET lacks multiple critical components of human visual search abilities, as also suggested by [27]. These include finite working memory, object recognition, and contextual reasoning, among others. Furthermore, the VGG16 backbone of eccNET constitutes only a first-order approximation to the intricacies of ventral visual cortex. Second, to convert fixations from eccNET to key press reaction time, we assumed that the relationship between number of fixations and RTs is independent of the experimental conditions. This assumption probably constitutes an oversimplification. While the approximate frequency of saccades tends to be similar across tasks, there can still be differences in the saccade frequency and fixation duration depending on the nature of the stimuli, on the difficulty of the task, on some of the implementation aspects such as monitor size and contrast, and even on the subjects themselves [52]. Third, the human psychophysics tasks involved both target present and target absent trials. Deciding whether an image contains a target or not (human psychophysics experiments) is not identical to generating a sequence of saccades until the target is fixated upon (model). It is likely that humans located the target when they pressed a key to indicate target presence, but this was not measured in the experiments and it is conceivable that a key was pressed while the subject was still fixating on a distractor. Fourth, humans tend to make "return fixations", whereby they fixate on the target, move the eyes away and come back to the target [52] while eccNET would stop at the first time of fixating on the target. Fifth, it is worth nothing that it is possible to obtain tighter quantitative fits to the RTs by incorporating a bottom-up saliency contribution to the attention map and fitting the parameters which corresponds to the relative contribution of top-down and bottom-up attention maps (see **Appendix H** and **Figures S20-21** for more details).

We introduce a biologically plausible visual search model that can qualitatively approximate the asymmetry of human visual search behaviors. The success of the model encourages further investigation of improved computational models of visual search, emphasizes the importance of directly comparing models with biological architectures and behavioral outputs, and demonstrates that complex human biases can be derived from model's architecture and the statistics of training images.

## Acknowledgements

This work was supported by NIH R01EY026025 and the Center for Brains, Minds and Machines, funded by NSF STC award CCF-1231216. Mengmi Zhang was supported by a postdoctoral fellowship of the Agency for Science, Technology and Research. Shashi Kant Gupta was supported by The Department of Biotechnology (DBT), Govt of India, Indo-US Science and Technology Forum (IUSSTF) and WINStep Forward under "Khorana Program for Scholars".

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
