# Supplementary Material for
# Visual Search Asymmetry:
# Deep Nets and Humans Share Similar Inherent Biases

Shashi Kant Gupta[1], Mengmi Zhang[2,3], Chia-Chien Wu[4], Jeremy M. Wolfe[4], and Gabriel Kreiman[2,3]

[1]Indian Institute of Technology Kanpur, India
[2]Children's Hospital, Harvard Medical School
[3]Center for Brains, Minds and Machines
[4]Brigham and Women's Hospital, Harvard Medical School
Address correspondence to gabriel.kreiman@tch.harvard.edu

## List of Supplementary Figures

35th Conference on Neural Information Processing Systems (NeurIPS 2021).

## A  Implementation Details of Six Psychophysics Experiments

**Experiment 1: Curvature.** This experiment is based on [19]. There were two conditions in this experiment: 1. Searching for a straight line among curved lines (**Figure 1A**, left), and 2. Searching for a curved line among straight lines (**Figure 1A**, right). The search image was 11.3 x 11.3 degrees of visual angle (dva). Straight lines were 1.2 dva long and 0.18 dva wide. Curved lines were obtained from an arc of a circle of 1 dva radius, the length of the segment was 1.3 dva, and the width was 0.18 dva. Targets and distractors were randomly placed in a 6 x 6 grid. Inside each of the grid cells, the objects were randomly shifted so that they did not necessarily get placed at the center of the grid cell. The target and distractors were presented in any of the four orientations: -45, 0, 45, and 90 degrees. Three set sizes were used: 8, 16, and 32. There was a total of 90 experiment trials per condition, equally distributed among each of the set sizes.

**Experiment 2: Lighting Direction.** This experiment is based on [10]. There were two conditions in this experiment: 1. Searching for left-right luminance change among right-left luminance changes (**Figure 1B**, left). 2. Searching for top-down luminance change among down-top luminance changes (**Figure 1B**, right). The search image was 6.6 x 6.6 dva. The objects were circles with a radius of 1.04 dva. The luminance changes were brought upon by 16 different levels at an interval of 17 on a dynamic range of [0, 255]. The intensity value for the background was 27. Targets and distractors were randomly placed in a 4 x 4 grid. Inside each of the grid cells, the objects were randomly shifted. Three set sizes were used: 1, 6, and 12. There was a total of 90 experiment trials per condition, equally distributed among each of the set sizes.

**Experiments 3-4: Intersection.** This experiment is based on [17]. There were four different conditions: 1. Searching for a cross among non-crosses (**Figure 1C**, left). 2. Searching for a non-cross among crosses (**Figure 1C**, right). 3. Searching for an L among Ts (**Figure 1D**, left). 4. Searching for a T among Ls (**Figure 1D**, right). Each of the objects was enclosed in a square of size 5.5 x 5.5 dva. The width of the individual lines used to make the object was 0.55 dva. Non-cross objects were made from the same cross image by shifting one side of the horizontal line along the vertical. The search image spanned 20.5 x 20.5 dva. The objects were randomly placed in a 3 x 3 grid. Inside each of the grid cells, the objects were randomly shifted. The target and distractors were presented in any of the four orientations: 0, 90, 180, and 270 degrees. Three set sizes were used: 3, 6, and 9. There was a total of 108 experiment trials per condition, equally distributed among each of the set sizes.

**Experiments 5-6: Orientation.** This experiment is based on [18]. There were four different conditions: 1. Searching for a vertical straight line among 20-degrees-tilted lines (**Figure 1E**, left). 2. Searching for a 20-degree-tilted line among vertical straight lines (**Figure 1E**, right). 3. Searching for a 20-degree tilted line among tilted lines of angles -80, -60, -40, -20, 0, 40, 60, 80 (**Figure 1F**, left). 4. Searching for a vertical straight line among tilted lines of angles -80, -60, -40, -20, 20, 40, 60, 80 (**Figure 1F**, right). Each of the objects was enclosed in a square of size 2.3 x 2.3 dva. The lines were of length 2 dva and width 0.3 dva. The search image spanned 11.3 x 11.3 dva. Targets and distractors were randomly placed in a 4 x 4 grid. Inside each of the grid cells, the objects were randomly shifted. In the heterogeneous cases (Experiment 6), distractors were selected such that the proportions of individual distractor angles were equal. Four set sizes were used: 1, 4, 8, and 12. There was a total of 120 experiment trials per condition, equally distributed among each of the set sizes.

## B  Computing key press reaction time from number of fixations

The proposed computational model of visual search predicts a series of fixations. The psychophysics experiments 1-6 did not measure eye movements and instead report a *key press reaction time* (RT) indicating when subjects found the target. To compare the model output to RT, we used data from a separate experiment that measured both RT and eye movements (**Figure S1**, **Appendix C**). We assume that the key press RT results from a combination of time taken by fixations plus a motor response time. Therefore to calculate key press reaction times in milliseconds from the number of fixations we used the linear fit in Equation 1. Where, $RT$ = reaction time in milliseconds, $N$ =

number of fixations until the target was found, $\alpha$ = duration of a single saccade + fixation = constant, and $\beta$ = motor response time = constant.

$$RT = \alpha * N + \beta \qquad (1)$$

The value for constants $\alpha$ and $\beta$ were estimated using the linear least-squares regression method on the data obtained from the experiment (**Figure S1**): $\alpha = 252.36$ milliseconds/fixation and $\beta = 376.27$ milliseconds. The correlation coefficient was $0.95$ ($p < 0.001$). Here we assume that both $\alpha$ and $\beta$ are *independent of the actual experiment* and use the same constant values for all the experiments (see **Section 5** in the main text)

## C    Experiment to convert fixations to key press reaction times

The experiment mentioned in **Appendix B** is detailed here (**Figure S1**). In this experiment, subjects had to find a rotated letter T among rotated Ls. Observer's key press reaction time and eye fixations data were recorded by SMI RED250 mobile eye tracker with sample rate 250Hz. Both eyes were tracked during the experiment. Each of the letters was enclosed in a square of 1 dva x 1 dva. The width of the individual lines used to make the letter was 0.33 dva. The target and distractors were randomly rotated between $1°$ to $360°$. Two set sizes were used: 42 and 80. Letters were uniformly placed in a $6 \times 7$ (42 objects) and $8 \times 10$ (80 objects) grid. The separation between adjacent letters was 2.5 dva. The search image consisting of the target and distractors was placed on the center of the screen of size 22.6 dva x 40.2 dva. The search image was centered on the screen. Stimuli were presented on a 15" HP laptop monitor with a screen resolution of 1920 x 1080 at a viewing distance of 47 cm. There were 20 practice trials and 100 experimental trials for each of two set sizes. Two set sizes were intermixed across trials. Observers used a keyboard to respond whether there was or was not a letter T in the display. The experiments were written in MATLAB 8.3 with Psychtoolbox version 3.0.12 [4, 11, 13]. Twenty-two observers participated in the experiment. All participants were recruited from the designated volunteer pool. All had normal or corrected-to-normal vision and passed the Ishihara color screen test. Participants gave informed consent approved by the IRB and were paid $11/hour.

## D    Converting pixels to degrees of visual angle for eccNET

We map the receptive field sizes in units of pixels to units of degrees of visual angle (dva) using 30 pixels/dva. This value indirectly represents the "clarity" of vision for our computational model. Since we have a stride of 2 pixels at each pooling layer, the mapping parameter $\eta$ from pixel to dva decreases over layers, we have $\eta_3 = (30/2)$ pixels/dva, $\eta_6 = (30/4)$ pixels/dva, $\eta_{10} = (30/8)$ pixels/dva, $\eta_{14} = (30/16)$ pixels/dva, and $\eta_{18} = (30/32)$ pixels/dva. To achieve better downsampling outcomes, the average-pooling operation also includes the stride [7] defining the movement of downsampling location. We empirically set a constant stride to be 2 pixels for all eccentricity-dependent pooling layers.

## E    Estimation of RF vs Eccentricity plot in Deep-CNN models

We estimated the RF vs Eccentricity plot in Deep-CNN Models through an experimental procedure. At first, for simplification and faster calculation, we replaced all the convolutional layers of the architecture with an equivalent max-pooling layer of the same window size. The same window size ensures that the simplified architecture's receptive field will be the same as the original one. We then feed the network with a complete "black" image and then followed by a complete "white" image, and saves the neuron index, which shows higher activation for the white image compared to the black one. This gives the indexes of the neurons which show activity for white pixels. After this, we feed the model with several other images having a black background with a white pixel area. The white portion of the images was spatially translated from the center of the image towards the periphery. For each of the neurons which have shown activity for the white image, we check its activity for all of the translated input images. Based on this activity, we estimate the receptive field size and eccentricity for the neurons. For an example unit $j$, if it shows activity for input images having white pixels at 6 dva to 8 dva, we say the RF for the unit $j$ is 2 dva and its eccentricity is 7 dva.

## F  Visualization of acuity of the input image at successive eccentricity-dependent pooling layers

After applying a convolution operation, the raw features of the input images get transformed to another feature space. For visualization purpose, we removed all the convolutional layers from the model and only kept the pooling layers. Therefore, after each pooling operation, we obtain an image similar to the input image with different acuity depending on the pooling operations. The corresponding visualization for eccNET is shown in **Figure 3A**.

| Input image size = 1200 px X 1200 px | | | | | | | | | | | | | | | |
|---|---|---|---|---|---|---|---|---|---|---|---|---|---|---|---|
| **Layer 3** | | | | | | | | | | | | | | | |
| **Distance (px)** 60 | 75 | 90 | 105 | 120 | 135 | 150 | 165 | 180 | 195 | 210 | 225 | 240 | 255 | 270 | 285 |
| **Local RF (px)** 2 | 2 | 2 | 2 | 2 | 2 | 2 | 2 | 2 | 2 | 2 | 2 | 2 | 2 | 2 | 2 |
| **Layer 6** | | | | | | | | | | | | | | | |
| **Distance (px)** 30 | 38 | 46 | 54 | 62 | 70 | 78 | 86 | 94 | 102 | 110 | 118 | 126 | 134 | 142 | 150 |
| **Local RF (px)** 2 | 2 | 2 | 2 | 2 | 2 | 2 | 2 | 2 | 2 | 2 | 2 | 2 | 2 | 2 | 2 |
| **Layer 10** | | | | | | | | | | | | | | | |
| **Distance (px)** 16 | 20 | 24 | 28 | 32 | 36 | 40 | 44 | 48 | 52 | 56 | 60 | 64 | 68 | 72 | 76 |
| **Local RF (px)** 2 | 3 | 3 | 4 | 4 | 5 | 5 | 6 | 6 | 7 | 8 | 8 | 9 | 9 | 10 | 10 |
| **Layer 14** | | | | | | | | | | | | | | | |
| **Distance (px)** 8 | 10 | 12 | 14 | 16 | 18 | 20 | 22 | 24 | 26 | 28 | 30 | 32 | 34 | 36 | 38 |
| **Local RF (px)** 2 | 3 | 3 | 4 | 5 | 5 | 6 | 6 | 7 | 8 | 8 | 9 | 10 | 10 | 11 | 12 |
| **Layer 18** | | | | | | | | | | | | | | | |
| **Distance (px)** 4 | 5 | 6 | 7 | 8 | 9 | 10 | 11 | 12 | 13 | 14 | 15 | 16 | 17 | 18 | 19 |
| **Window size (px)** 2 | 3 | 3 | 4 | 5 | 5 | 6 | 6 | 7 | 8 | 8 | 9 | 10 | 10 | 11 | 12 |

Table S1: **Variations of local receptive field size vs distance from the fixation at each pooling layer of eccNET.** This table shows the variations in local receptive field size ($r_l^j$) with the change in distance from the fixation ($d_{l+1}^j$) with respect to the output of the pooling layer at layers 3, 6, 10, 14, and 18 of eccNET. The values are in pixels (px).

## G  Training details of VGG16 from scratch on rotated ImageNet and MNIST datasets

### G.1  Rotated ImageNet

The training procedure follows the one in the original VGG16 model [15]. The images were preprocessed according to original VGG16 model, i.e., resized such that all the training images had a size of 256 by 256 pixels. Then a random crop of 224 x 224 was taken. Then the image was flipped horizontally around the vertical axis. After this the image was rotated by 90 degrees in the anti-clockwise direction. The VGG16 architecture was built using TensorFlow Keras deep learning library [1], with the same configuration as the original model. Training was carried out by minimizing the categorical cross-entropy loss function with the Adam optimiser [9], with initial learning rate of $1 \times 10^{-4}$, step decay of $1 \times 10^{-1}$ at epoch 20. The training was regularised by weight decay of $1 \times 10^{-5}$. The training batch size was 150. The learning rates were estimated using the LR range finder technique [16]. The training was done until 24 epochs, which took approximately 1.5 days on 3 NVIDIA GeForce RTX 2080 Ti Rev. A (11GB) GPUs.

### G.2  MNIST

The procedure is the same as in the previous section. The training batch size here was 32 and training was done for 20 epochs.

# H A better quantitative fit to reaction time plots by using bottom-up saliency and performing parameter fitting

In the main paper, it is important to emphasize that we did not fit any parameter in the model to the behavioral data in the results shown in the paper. Given the lack of parameter tuning, and the multiple differences between humans and machines (e.g., how humans are "trained", target-absent trials only for humans, motor cost for humans, target localization versus detection), one may not expect precise fitting of reaction times. What we find remarkable is that even without such parameter tuning, it is possible to capture fundamental properties of human behavior. We consider the results to demonstrate as a proof-of-principle, that neural network models can show the type of asymmetric properties that are evident for humans without doing any task specific training or fitting parameters to capture those asymmetry.

If we allow ourselves to do parameter fitting to specific search asymmetry experiments and also include a bottom-up saliency model, it is possible to obtain tighter quantitative fits to the reaction times as well as capture the asymmetry in case of **Experiment E** where the eccNET model initially failed. We argue that in the case of **Experiment E**, it is quite possible that bottom-up saliency is playing a major role in driving asymmetry in humans which is consistent with the ideas from some psychophysics studies [14, 6, 8].

**Bottom-up saliency model (eccNET$_{bu}$)**

The bottom-up saliency model is based on the information maximization approach (Figure S20). This method has been previously shown to be effective to find salient regions in an image ([5]). The original implementation used a representation based on independent component analysis. Instead, here we used the feature maps extracted from the computational model of the visual cortex (eccNET). At layer $l$ of eccNET, we extracted feature maps of size $C_l \times H_l \times W_l$, where $C_l$ is the number of channels. and $H_l$, $W_l$ denote the height and width, respectively. On the $c^{th}$ channel of the feature maps, we define the histogram function $F_{l,c,n}(\cdot)$, which takes the activation values $y^j_{l,c,n}$ as inputs and outputs its corresponding frequency among all individual units $j$ at all $H_l \times W_l$ locations at the $n^{th}$ fixation. Next, the model calculates the probability distribution for each unit $j$ on the $c^{th}$ feature map at layer $l$ and $n^{th}$ fixation:

$$p^j_{l,c,n} = \frac{F_{l,c,n}(y^j_{l,c,n})}{\sum_{i=0,1,...,W_l \times H_l} F_{l,c,n}(y^i_{l,c,n})} \tag{2}$$

where $p^j_{l,c,n}$ denotes how prevalent the activation value $y^j_{l,c,n}$ is over all units $j$ on the $c^{th}$ channel feature map. To capture attention drawn to less frequent visual features on an image, the model uses the normalized negative log probability to compute a saliency map for each channel and then averages the saliency maps over all channels and then over all selected layers $l = 9, 13, 17$ to output the overall saliency map $S_n$ at the $n$th fixation:

$$S_n = \sum_{l=9,13,17} \sum_c^{C_l} \frac{-\log(p^j_{l,c,n})}{p_{max} - p_{min}} \tag{3}$$

Where:

$$p_{max} = \max(\{-\log(p^i_{l,c,n}) : i = 1, 2, ..., H_l \times W_l\})$$
$$p_{min} = \min(\{-\log(p^i_{l,c,n}) : i = 1, 2, ..., H_l \times W_l\})$$

where the normalization of negative log probability is carried out by taking the difference between the maximum and minimum negative log probability among all the individual units $i$ in the $c$th channel at layer $l$. Since not all feature maps at the selected layers are of the same size, we downsampled individual saliency maps in the lower layers $l = 9, 13$ to be of the same size as those at layer $l = 17$.

**Integration of bottom-up and top-down maps**

Given the overall saliency map $S_n$ and the overall top-down activation map $A_n$ at the $n$th fixation, we normalize both maps within [0,1] and compute the overall attention map $O_n$ as a weighted linear combination of both maps. $w_{S,n}$ and $w_{A,n}$ denotes the weights applied on the bottom-up saliency

map $S_n$ and the top-down modulation map $A_n$ respectively. These coefficients control the relative contribution of bottom-up and top-down attention at each fixation. Now, we can simply use some 10-15 examples from each search experiments to fit the parameter $w_{S,n}$ to get a better quantitative fit. To further eliminate overfitting problem, we clustered these search tasks into three groups and we proposed three corresponding decision bias schemes for individual group of experiments: scheme (1) no saliency, scheme (2) equal saliency and top down, scheme (3) strong saliency. These three schemes effect the decision bias only at the first and second fixation ($n = 1, 2$) in each individual trial. For the subsequent fixations ($n > 2$), we argued that humans are strongly guided by top-down modulation effect with minimal bottom-up effect; that is $w_{S,n} = 0$ and $w_{A,n} = 1$ for all $n > 2$ regardless of the nature of visual search experiments. We formulated the computation of overall attention map as follows:

$$O_n = w_{S,n}S_n + w_{A,n}A_n \tag{4}$$

where

$$\begin{cases} w_{S,n} = 0, w_{A,n} = 1 & \text{if scheme (1) and } n = 1 \\ w_{S,n} = 0.5, w_{A,n} = 0.5 & \text{if scheme (2) and } n = 1 \\ w_{S,n} = 1, w_{A,n} = 0 & \text{if scheme (3) and } n = 1 \\ w_{S,n} = 0, w_{A,n} = 1 & \text{if scheme (1) and } n = 2 \\ w_{S,n} = 0.37, w_{A,n} = 0.63 & \text{if scheme (2) and } n = 2 \\ w_{S,n} = 0.37, w_{A,n} = 0.63 & \text{if scheme (3) and } n = 2 \\ w_{S,n} = 0, w_{A,n} = 1 & \text{if } n > 2 \end{cases} \tag{5}$$

Search tasks belonging to scheme (1) are Line among Curves (**Figure 1A**), Curve among Lines (**Figure 1A**), Cross among No-Intersections (**Figure 1C**), and No-Intersection among Crosses (**Figure 1C**). Search tasks belonging to scheme (2) are L among Ts (**Figure 1D**), T among Ls (**Figure 1D**), and Orientation Heterogeneous T 20 (**Figure 1F**). The rest of the task belongs to scheme (3). The results after introducing these changes are shown in **Figure S21**

## I Integration of attention maps in eccNET

Here we follow the details of calculation of the weight coefficients to merge the attentional maps (described in Section 3, below Equation 2 in the main text). The following numbers are taken from the example illustrated in **Figure S22**.

$$\begin{aligned} W_1 &= \frac{max(A_1)}{\sum_{i=1}^{i=3} max(A_i)} = \frac{415261}{415261 + 164618 + 17118} = 0.696 \\ W_2 &= \frac{max(A_2)}{\sum_{i=1}^{i=3} max(A_i)} = \frac{164618}{415261 + 164618 + 17118} = 0.276 \\ W_3 &= \frac{max(A_3)}{\sum_{i=1}^{i=3} max(A_i)} = \frac{17118}{415261 + 164618 + 17118} = 0.029 \end{aligned} \tag{6}$$

Here is the step-by-step calculation for point **P1**:

$$\begin{aligned} A_1(489, 69) &= 412454 \implies A_{1,normalized}(489, 69) = \frac{A_1(489, 69) - min(A_1)}{max(A_1) - min(A_1)} \\ &= \frac{412454 - 255590}{415261 - 255590} = 0.982 \end{aligned} \tag{7}$$

$$\begin{aligned} A_2(489, 69) &= 143382.031 \implies A_{2,normalized}(489, 69) = \frac{A_2(489, 69) - min(A_2)}{max(A_2) - min(A_2)} \\ &= \frac{143382 - 57584}{164618 - 57584} = 0.802 \end{aligned} \tag{8}$$

$$A_3(489, 69) = 9491.743 \implies A_{3,normalized}(489, 69) = \frac{A_3(489, 69) - min(A_3)}{max(A_3) - min(A_3)}$$

$$= \frac{9492 - 846}{17118 - 846} = 0.531 \tag{9}$$

Therefore, the value of point $P_1$ in the overall map ($O_m$) will be:

$$O_m(489, 69) = \sum_{i=1}^{i=3} W_i * A_{i,normalized} = 0.982 * 0.696 + 0.802 * 0.276 + 0.531 * 0.029 = 0.92 \tag{10}$$

And the value of point $P_1$ in the overall map without any normalization and scaling ($O_{m,ns}$) will be:

$$O_{m,ns}(489, 69) = \sum_{i=1}^{i=3} A_i = 412454 + 143382 + 9492 = 565328 \tag{11}$$

We follow a similar procedure for point **P2**.

$$A_1(108, 427) = 392635 \implies A_{1,normalized}(108, 427) = \frac{392635 - 255590}{415261 - 255590} = 0.858 \tag{12}$$

$$A_2(108, 427) = 163745 \implies A_{2,normalized}(108, 427) = \frac{163745 - 57584}{164618 - 57584} = 0.992 \tag{13}$$

$$A_3(108, 427) = 13075 \implies A_{3,normalized}(108, 427) = \frac{13075 - 846}{17118 - 846} = 0.752 \tag{14}$$

$$O_m(108, 427) = 0.858 * 0.696 + 0.992 * 0.276 + 0.752 * 0.029 = 0.892 \tag{15}$$

$$O_{m,ns}(108, 427) = 392635 + 163745 + 13075 = 569455 \tag{16}$$

## J    Perfect match of eccentricity-dependent sampling to the macaque data

There is certainly ample room to build better approximations of the receptive field sizes to create a perfect match of eccentricity-dependent sampling to the macaque data. But, it is worth noting that we are not aiming for a perfect quantitative match with macaque data; but for preserving the trend of eccentricity versus receptive field sizes.

It is worth pointing out that the curves shown for macaques, as reproduced in **main Figure 2B right**, constitute average measurements. There is considerable variation in the receptive field sizes, even at a fixed eccentricity and fixed visual area. As one example of many, consider the variation in [12].

It is also worth pointing out that there are extensive measurements of receptive field sizes of individual neurons in macaque monkeys (and also cats and rodents), but there is essentially no such measurement for humans. There exist field potential measurements of receptive fields in humans (e.g. [20] for early visual areas and [2] for higher visual areas). Thus, even if we strived to make a better fit to the average macaque data, it is not very clear that this would help us better understand the behavioural measurements in this study which were conducted in humans.

Furthermore, there are various constraints and computational limits for making a perfect fit:

1. Images are represented as "quantised pixel units", i.e., we have limited pixel sizes to use.

2. Scaling the input image size can be done to map some fractional window size of "0.5x0.5" equivalent to some integral window size of "2x2" or "3x3". But this comes at the cost of using a large size of the input image. There's a memory limitation on the GPU front on how large the images we can use are.

3. In principle, we could use interpolation between the neighbouring pixels while applying the pooling operation but we have not tried this.

Thus, for current study we did not focused on creating a perfect match which allowed us in making the design simplistic, with two parameters (slope of eccentricity versus receptive field sizes $\gamma$, scaling factor converting degrees of visual angle to pixels $\eta$)

**A.**

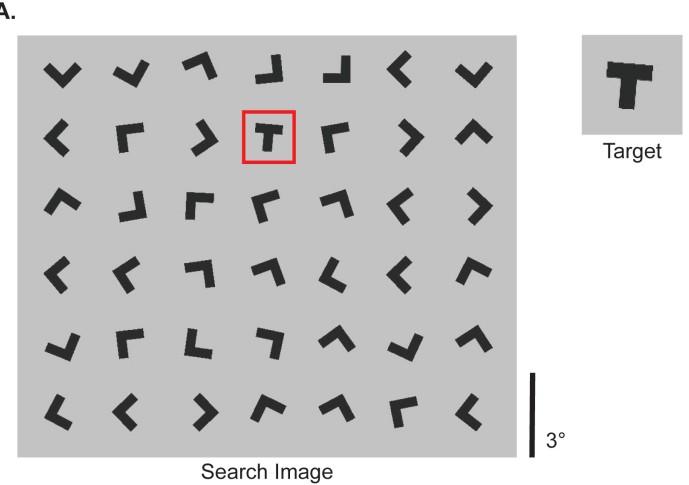

Search Image

Target

3°

**B.**

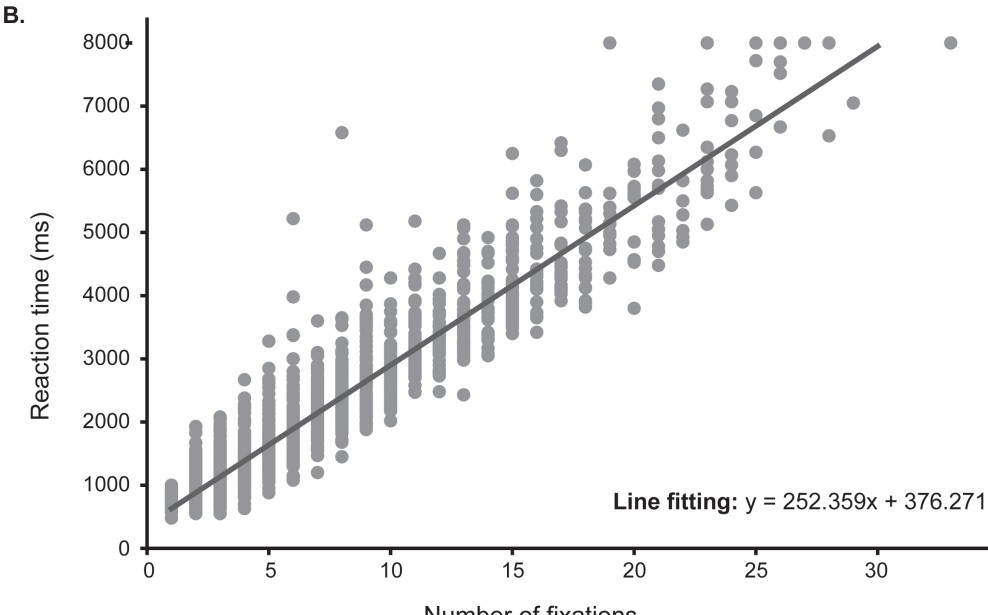

**Line fitting:** y = 252.359x + 376.271

Number of fixations

Figure S1: **Experiment to model reaction time from number of fixations. A.** Example from the T vs L visual search task used to evaluate the relationship between reaction times and number of fixations. **B.** Reaction time grows linearly with the number of fixations. Each gray point represents a trial. A line was fit to these data: $R(ms) = \alpha * n + \beta$. A fit using linear least square regression gave $\alpha = 252.359$ ms/fixation and $\beta = 376.271$ ms ($r^2 = 0.90$, $p < 0.001$). This linear fit was used throughout the manuscript to convert the number of fixations in the model to reaction time in milliseconds for comparison with human data.

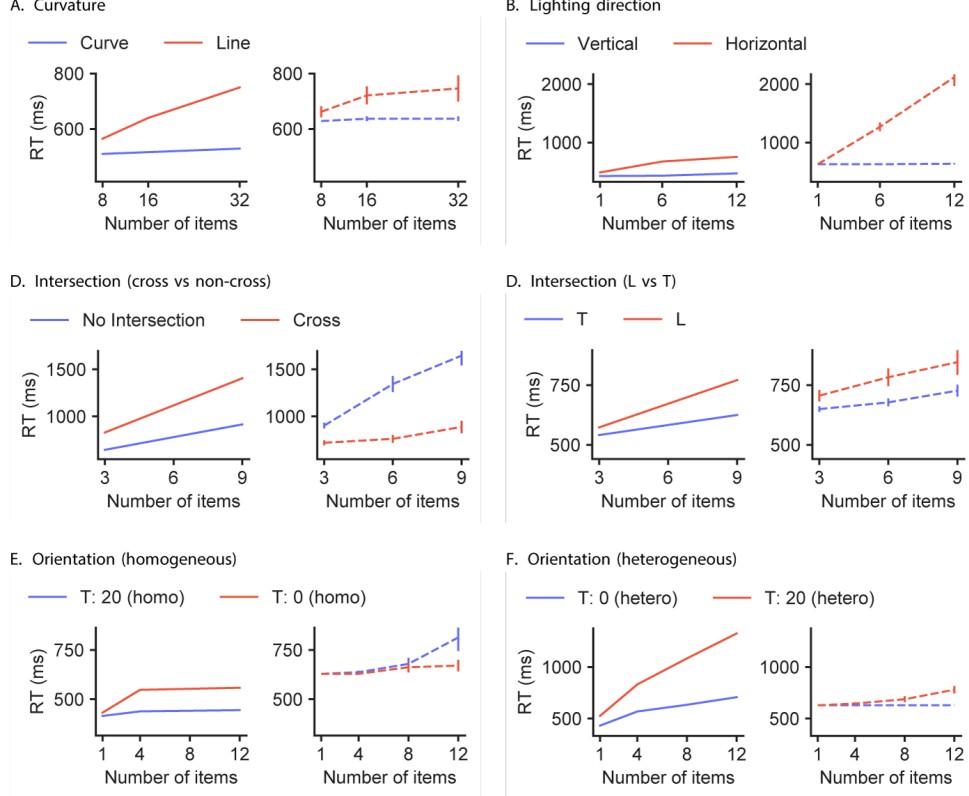

Figure S2: **IVSN - Reaction time as a function of the number of objects in the display for each of the six experiments for the IVSN model [21]**. The figure follows the format of **Figure 4** in the main text.

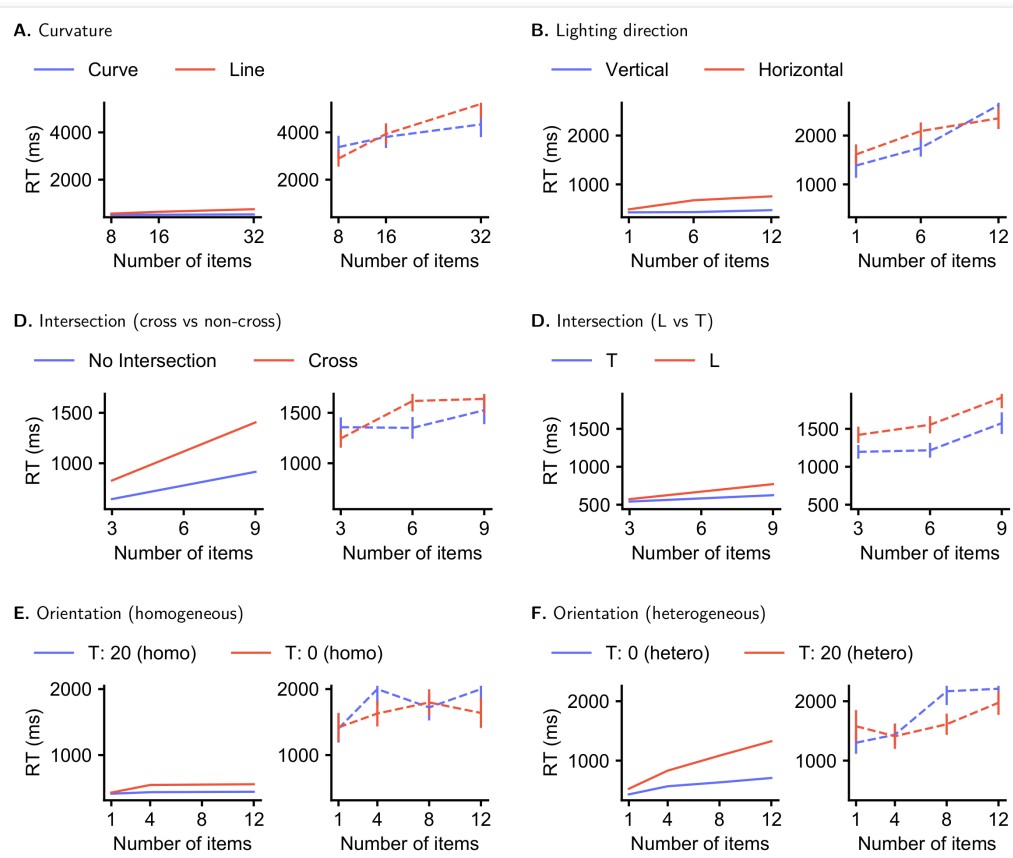

Figure S3: **Chance - Reaction time as a function of the number of objects in the display for each of the six experiments for the chance model**. The figure follows the format of **Figure 4** in the main text.

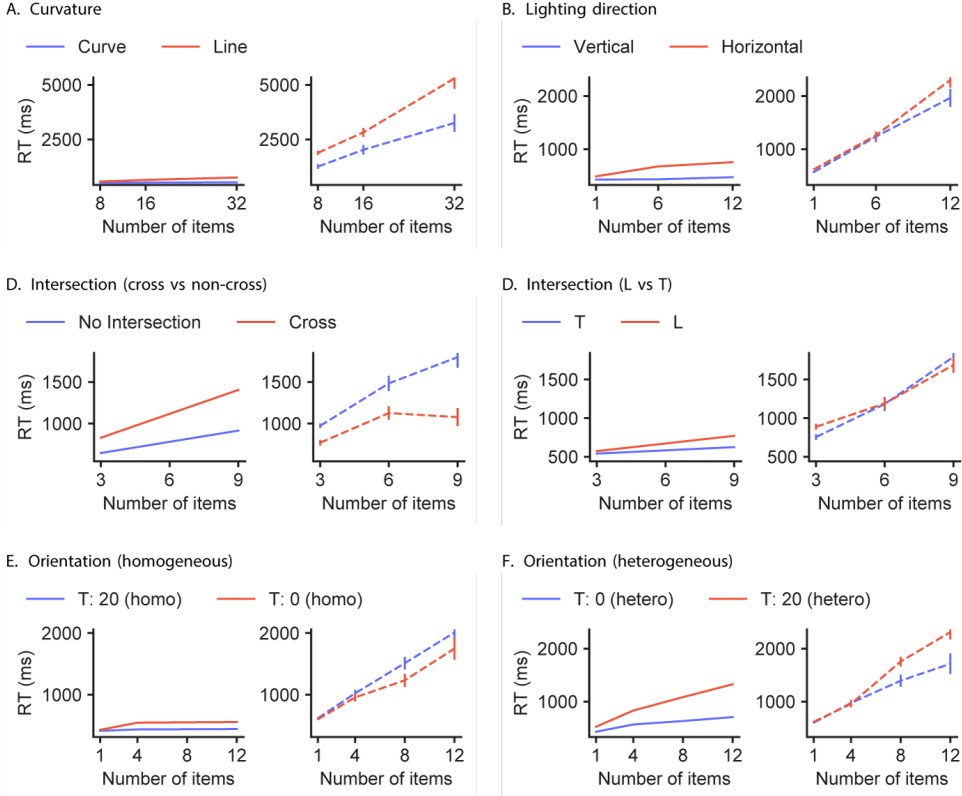

Figure S4: **GBVS - Reaction time as a function of the number of objects in the display for each of the six experiments for the bottom-up saliency model**. The figure follows the format of **Figure 4** in the main text.

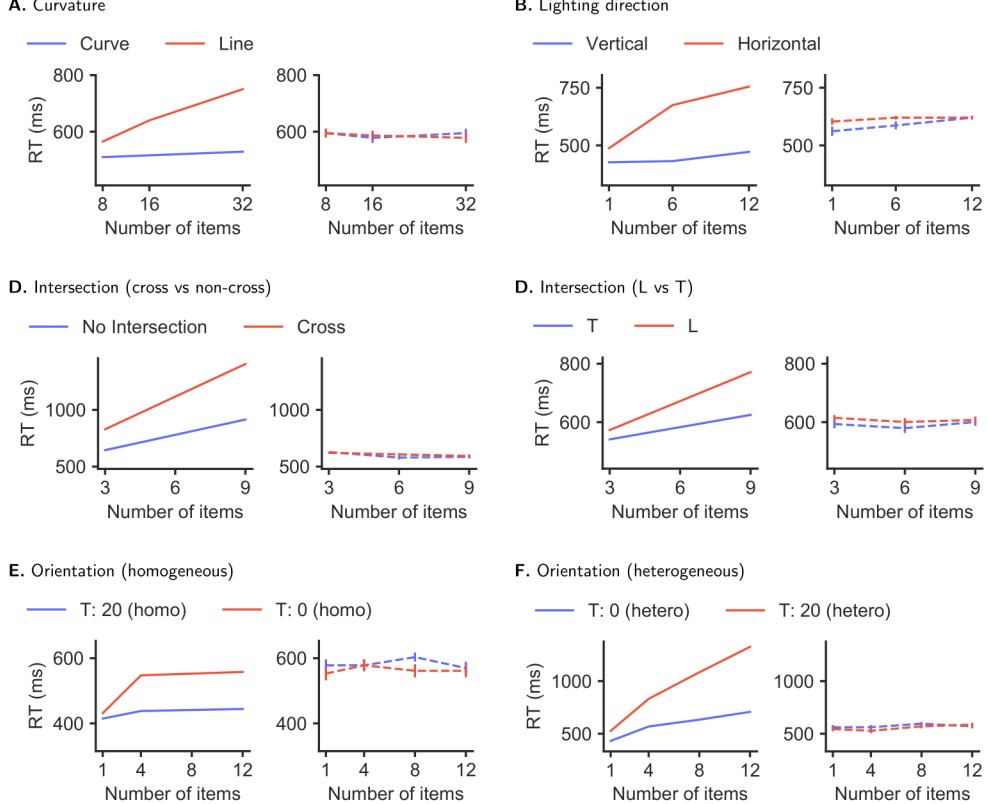

Figure S5: **pixelMatch - Reaction time as a function of the number of objects in the display for each of the six experiments for the template-matching model**. The figure follows the format of **Figure 4** in the main text.

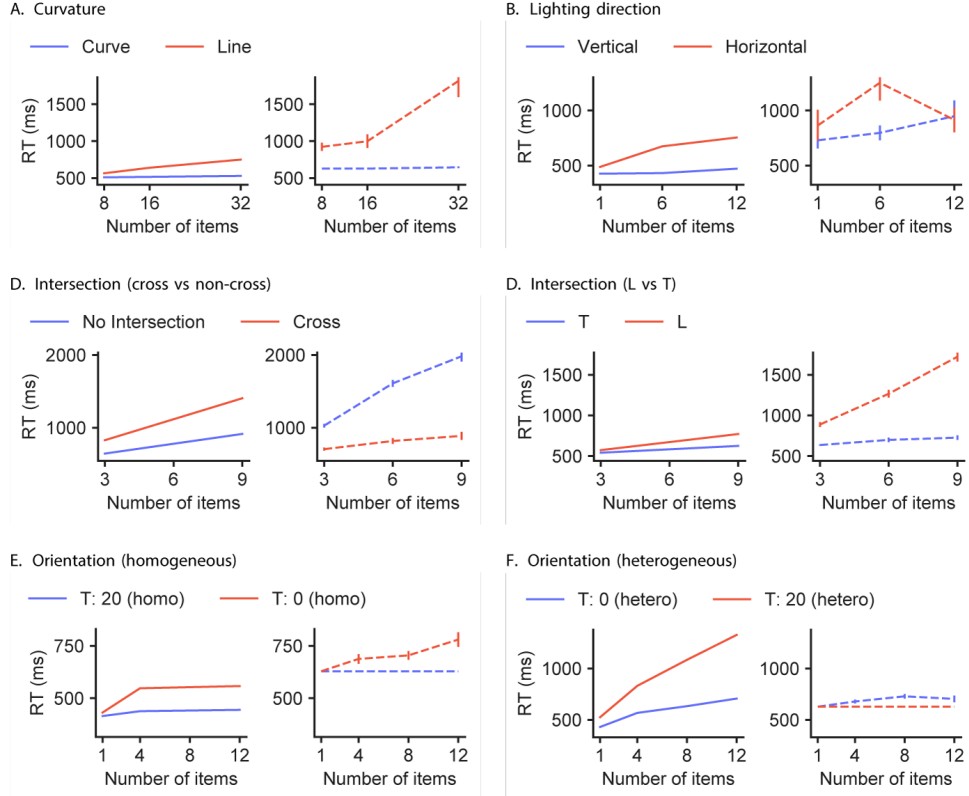

Figure S6:  **eccNET**$_{noecc}$**- Reaction time as a function of the number of objects in the display for each of the six experiments for the eccNET model without eccentricity-dependent sampling**. The figure follows the format of **Figure 4** in the main text.

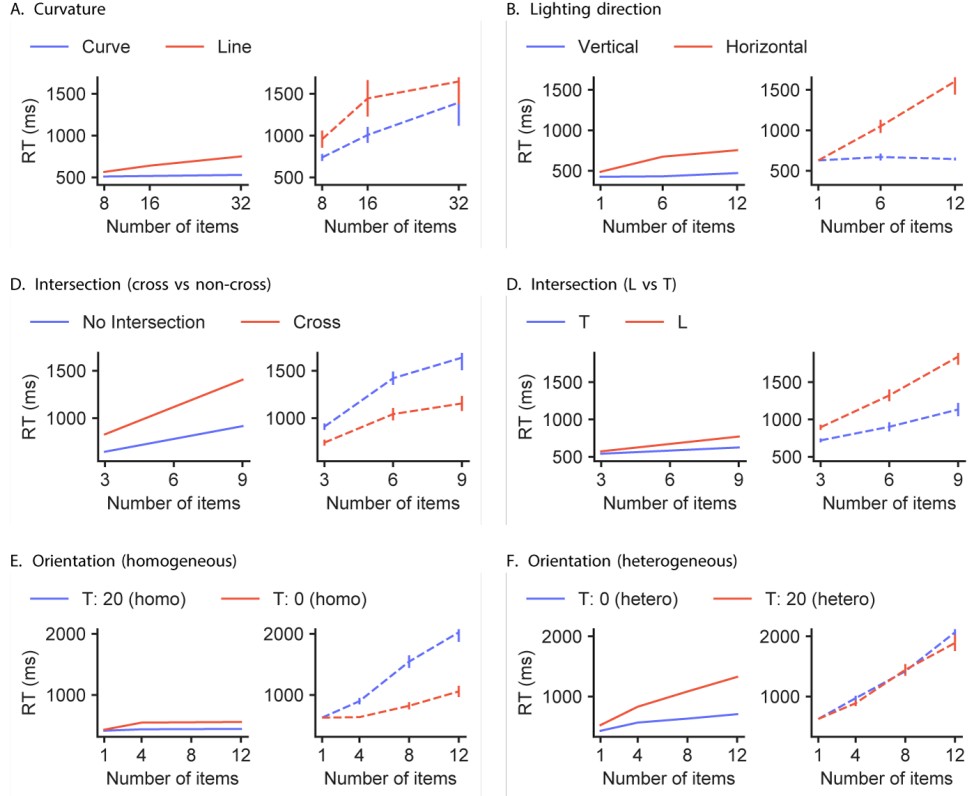

Figure S7: **eccNET**$_{18 \to 17}$ **- Reaction time as a function of the number of objects in the display for each of the six experiments for the eccNET model using top-down modulation only at the top layer**. The figure follows the format of **Figure 4** in the main text.

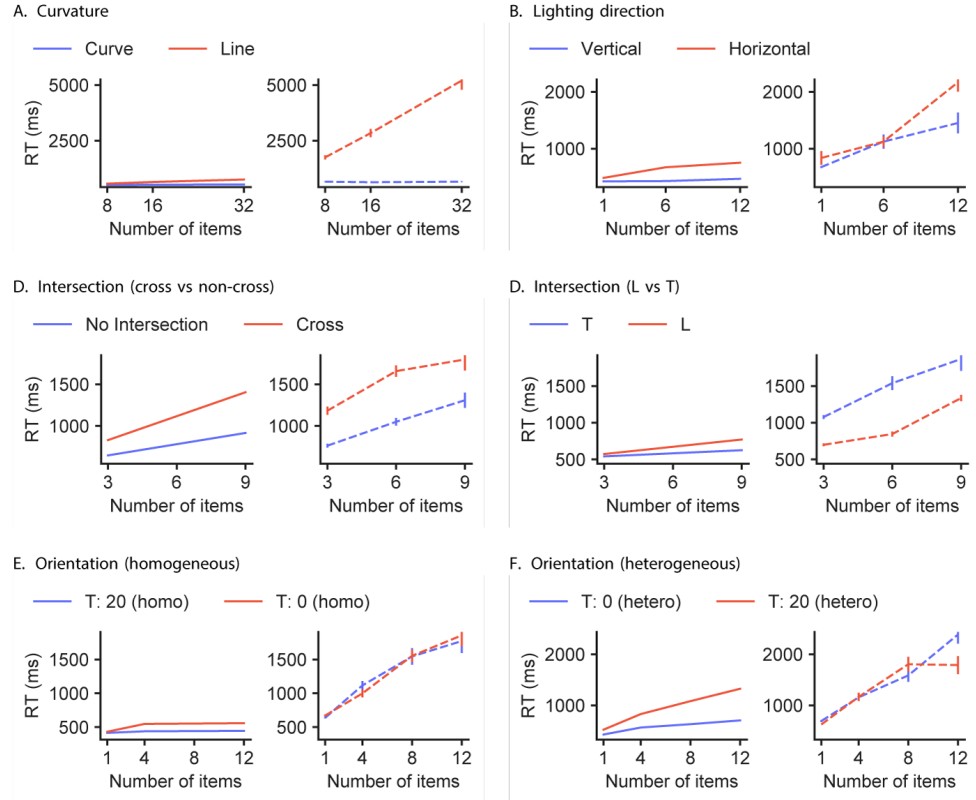

Figure S8: **eccNET**$_{MNIST}$ **- Reaction time as a function of the number of objects in the display for each of the six experiments for eccNET trained with the MNIST dataset**. The figure follows the format of **Figure 4** in the main text.

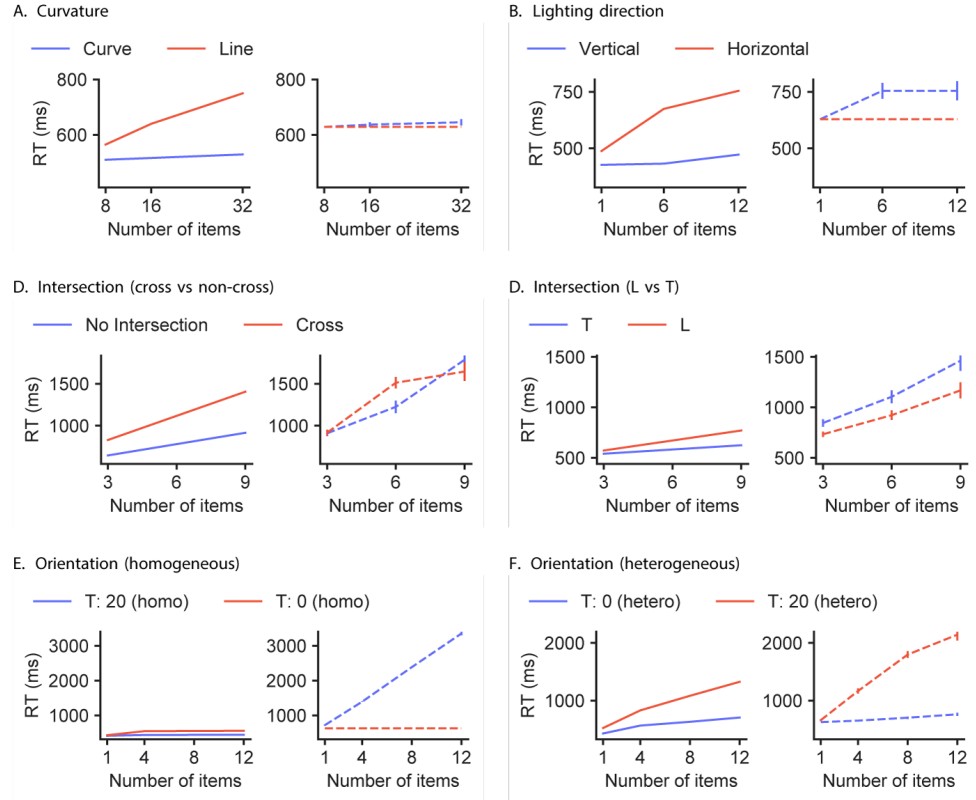

Figure S9: **eccNET**$_{Rot90}$ **- Reaction time as a function of the number of objects in the display for each of the six experiments for the eccNET model trained on a 90-degree rotated version of ImageNet**. The figure follows the format of **Figure 4** in the main text.

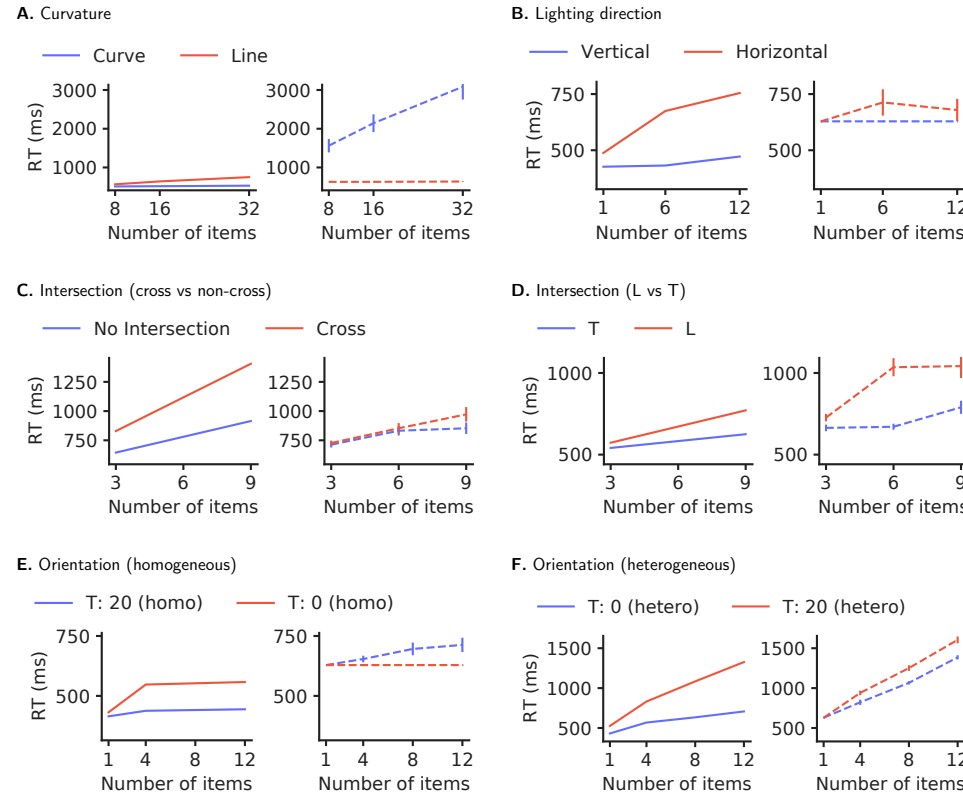

Figure S10: **eccNET**$_{fisheye}$ **- Reaction time as a function of the number of objects in the display for each of the six experiments for the eccNET model trained on fisheye distorted images of ImageNet**. The figure follows the format of **Figure 4** in the main text.

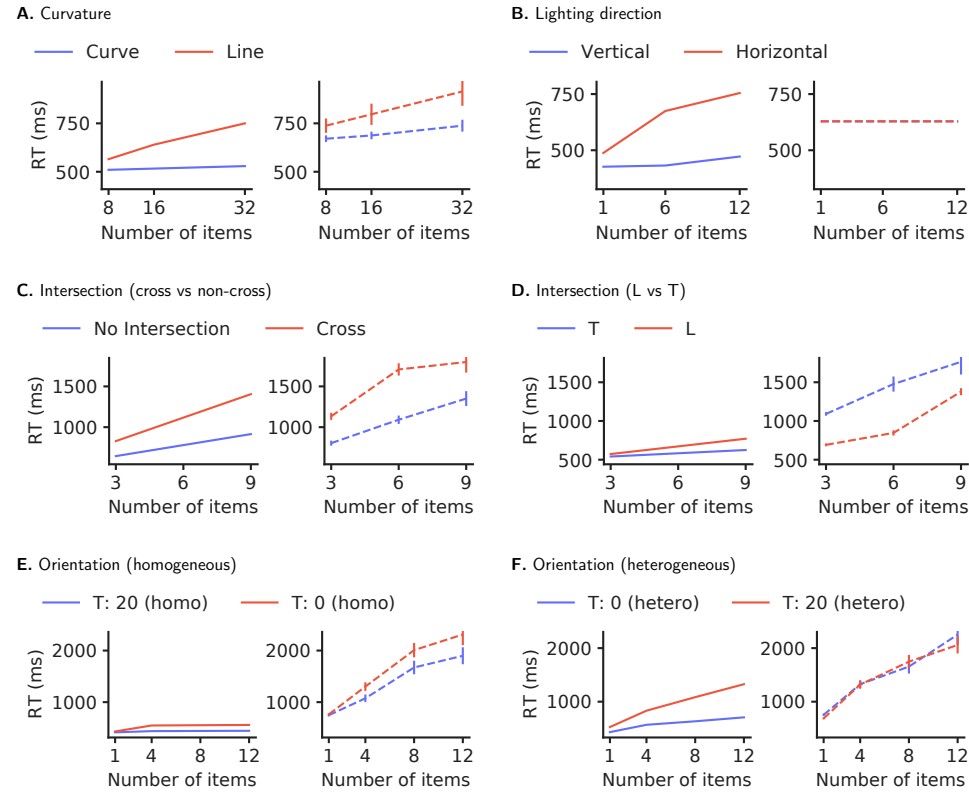

Figure S11: **eccNET**$_{lines}$ **- Reaction time as a function of the number of objects in the display for each of the six experiments for the eccNET model trained on images having grids formed using straight lines**. The figure follows the format of **Figure 4** in the main text.

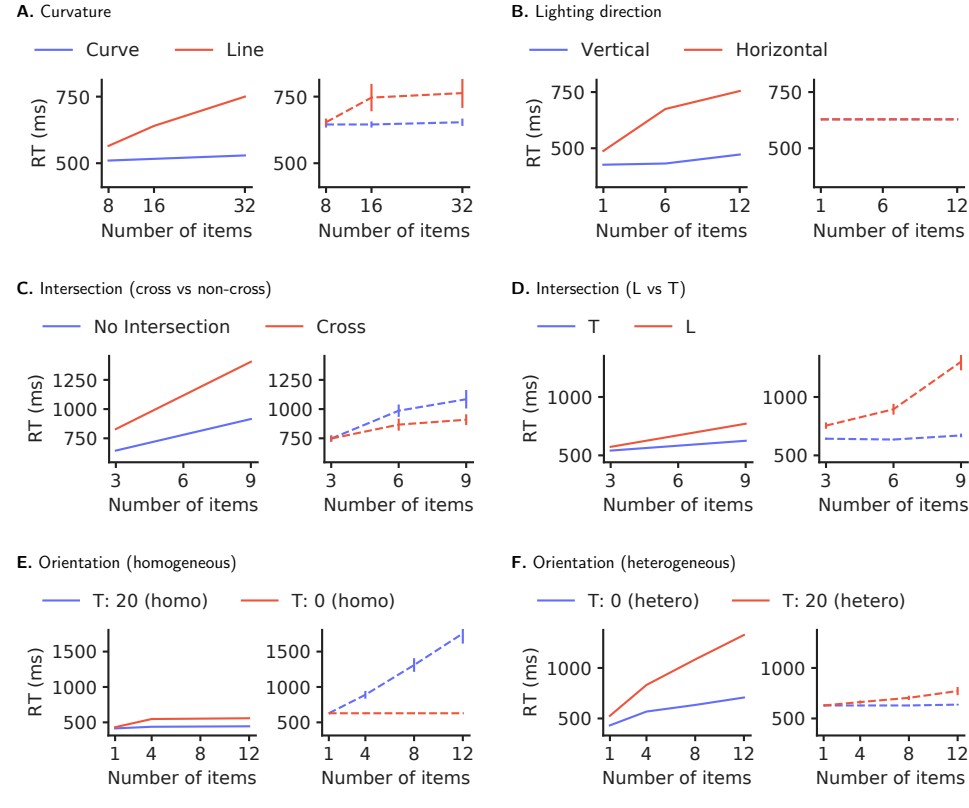

Figure S12: **eccNET**$_{places365}$ **- Reaction time as a function of the number of objects in the display for each of the six experiments for the eccNET model trained on places 365 dataset**. The figure follows the format of **Figure 4** in the main text.

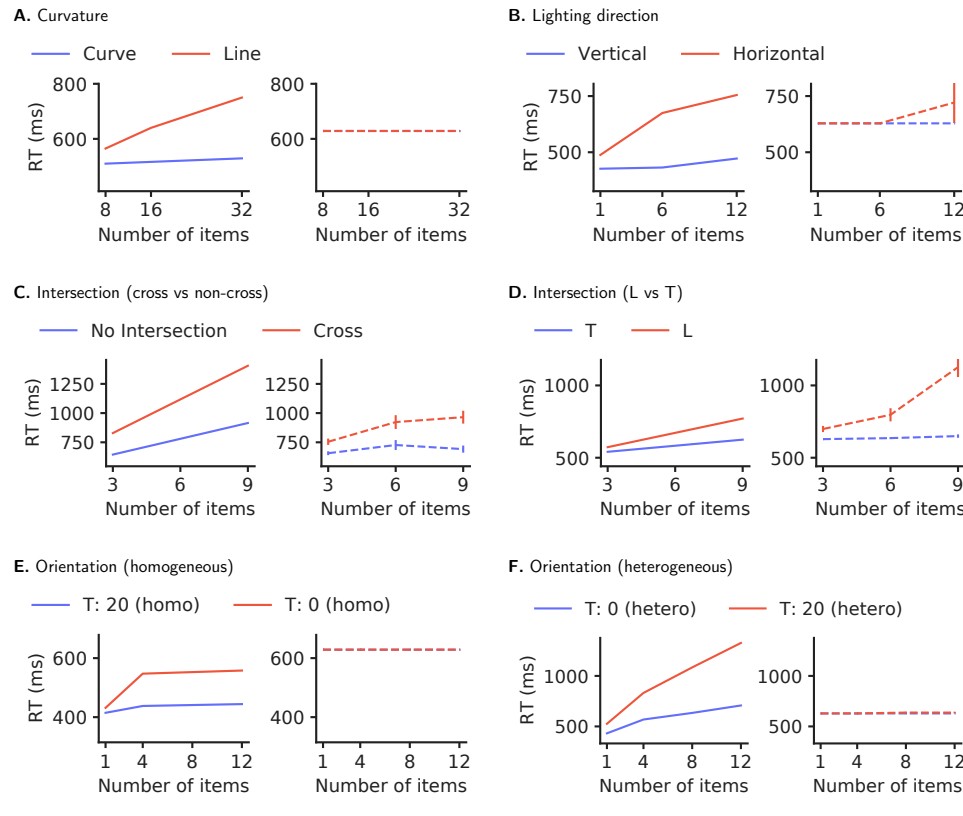

Figure S13: **eccNET**$_{places365,Rot90}$ **- Reaction time as a function of the number of objects in the display for each of the six experiments for the eccNET model trained on a 90-degree rotated version of places 365 dataset**. The figure follows the format of **Figure 4** in the main text.

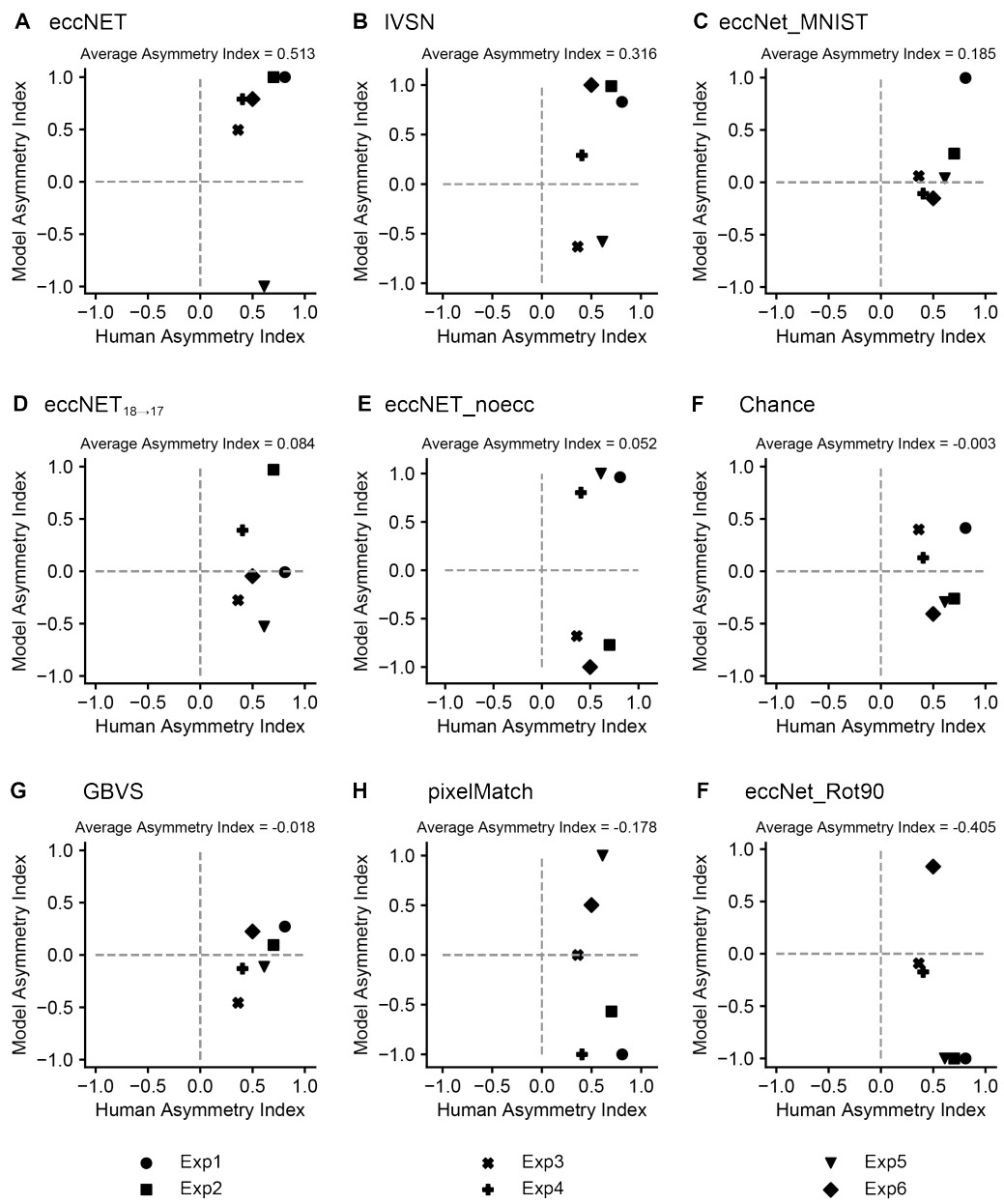

Figure S14: **Asymmetry Index for other models vs humans**. The figure follows the format of **Figure 5A** in the main text. Part **A** of this figure reproduces **Figure 5A** for comparison purposes.

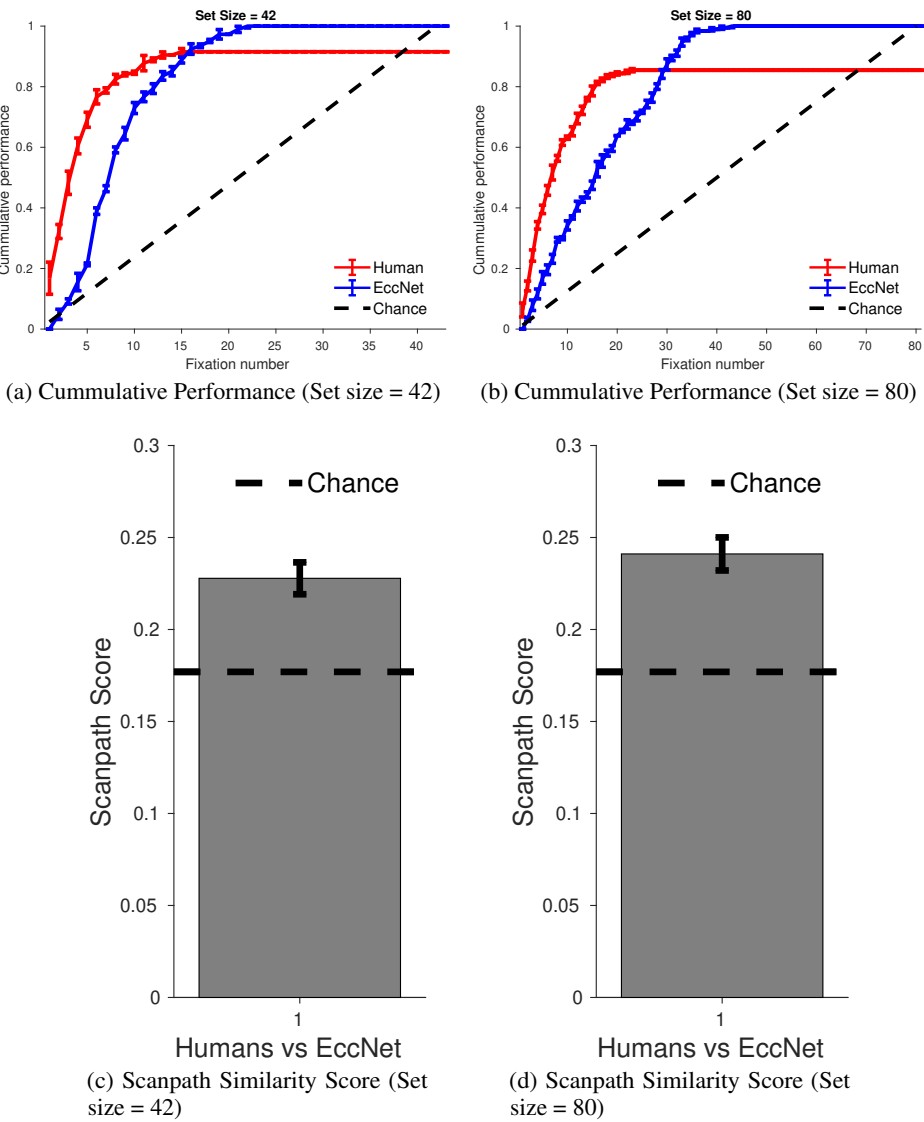

Figure S15: **Human-model eye movement comparison in T VS L experiment**. (a-b) Cumulative performance as a function of fixation number for humans (red), EccNet (blue), and chance model (dashed line). Error bars denote SEM, n = 5 subjects. Depending on the number of items on the search array, (a) shows the set size of 42 and (b) shows the set size of 80. (c-d) Image-by-image consistency in the spatiotemporal pattern of fixation sequences when the set size is 42 (c) and 80 (d).

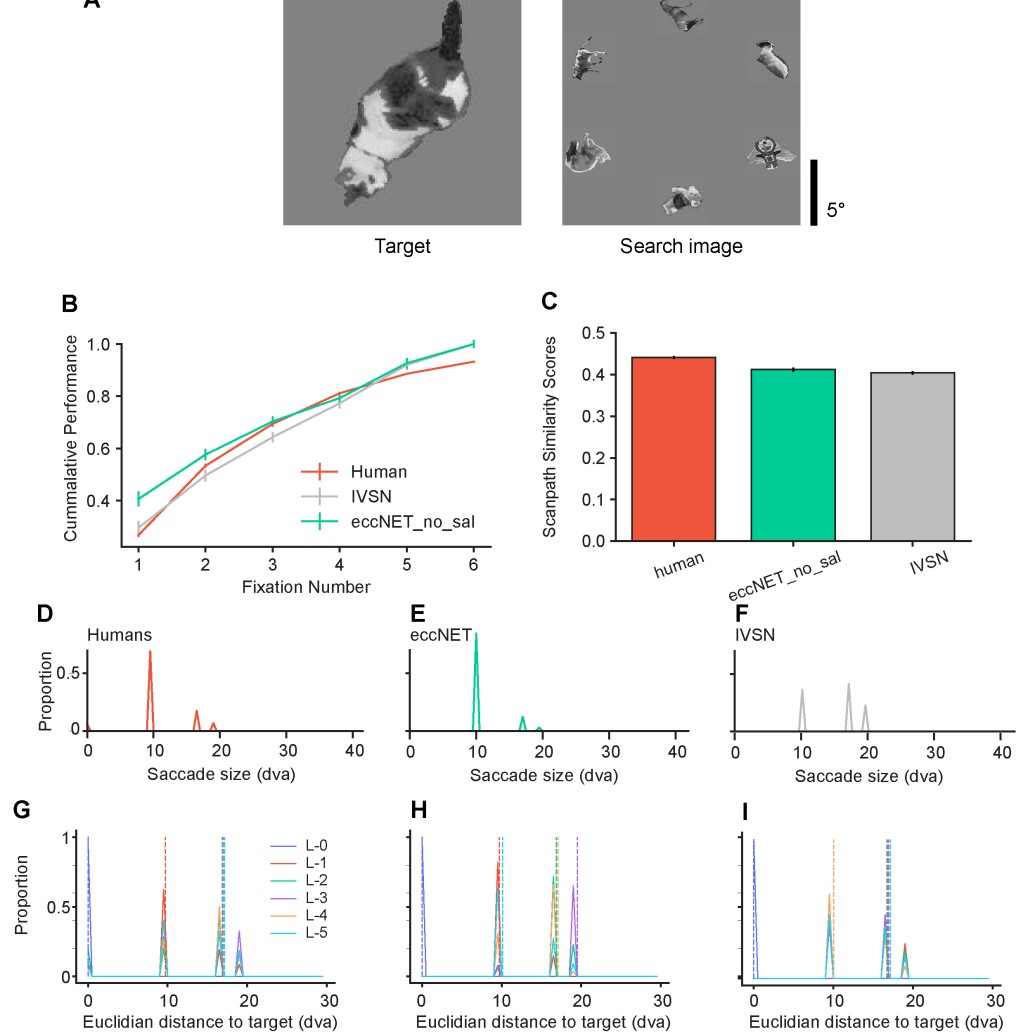

Figure S16: **The eccNET model matches previous visual search experiments with object arrays** ([21]). **A.** Example target and search images **B.** Cumulative search performance as a function of fixation number for humans (red), eccNET (green) and IVSN (gray). IVSN is the model proposed in [21]. **C.** Scanpath similarity scores between humans (red), between humans and ECCnet (green), and between humans and IVSN (gray). The scanpath similarity score measures the similarity between two two eye movement sequences ([3, 21]). **D-F.** Distribution of saccade sizes. **G-I**. Distribution of Euclidean distance from target location to either of the last six fixation locations.

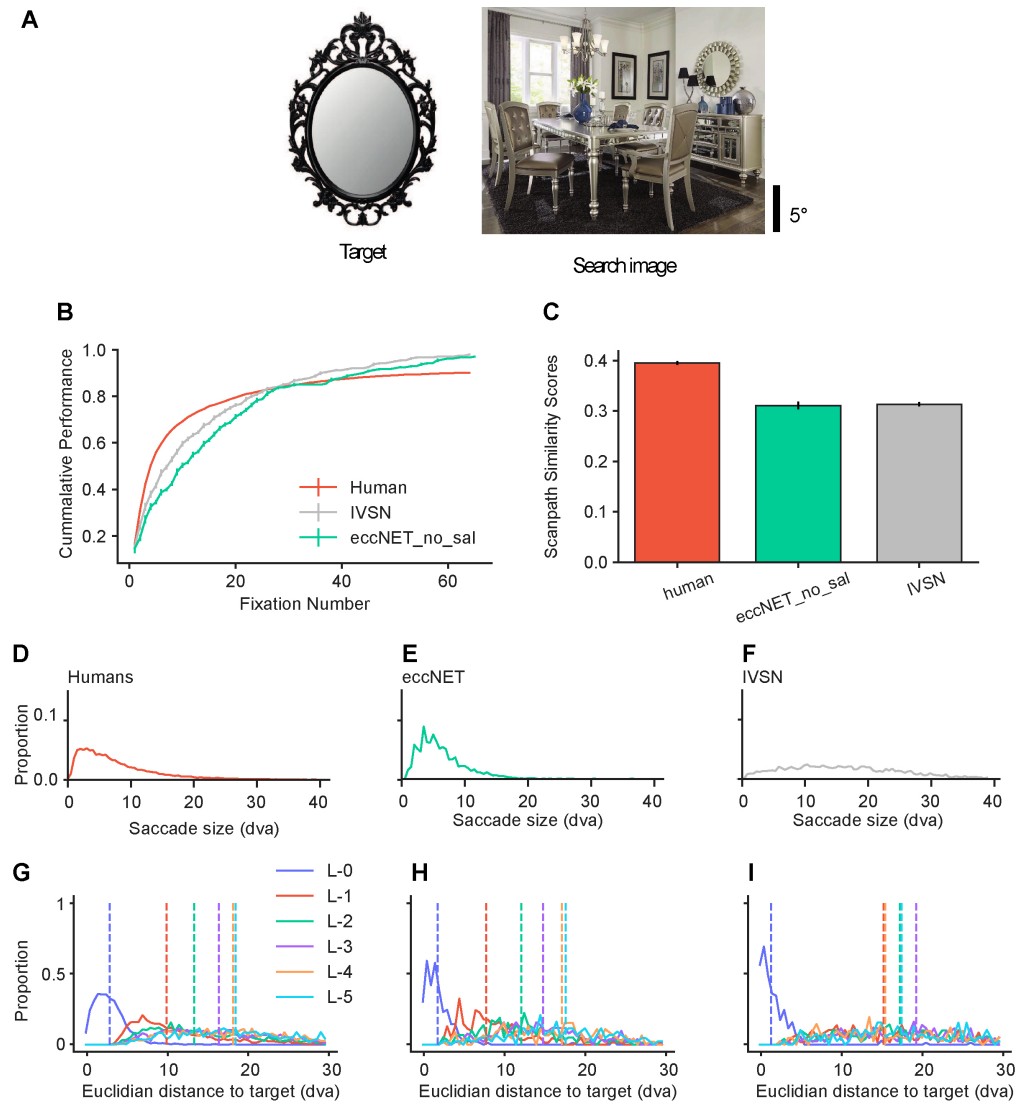

Figure S17: **The eccNET model matches previous visual search experiments with natural images** ([21]). The format and conventions in this figure are the same as in **Figure S16**.

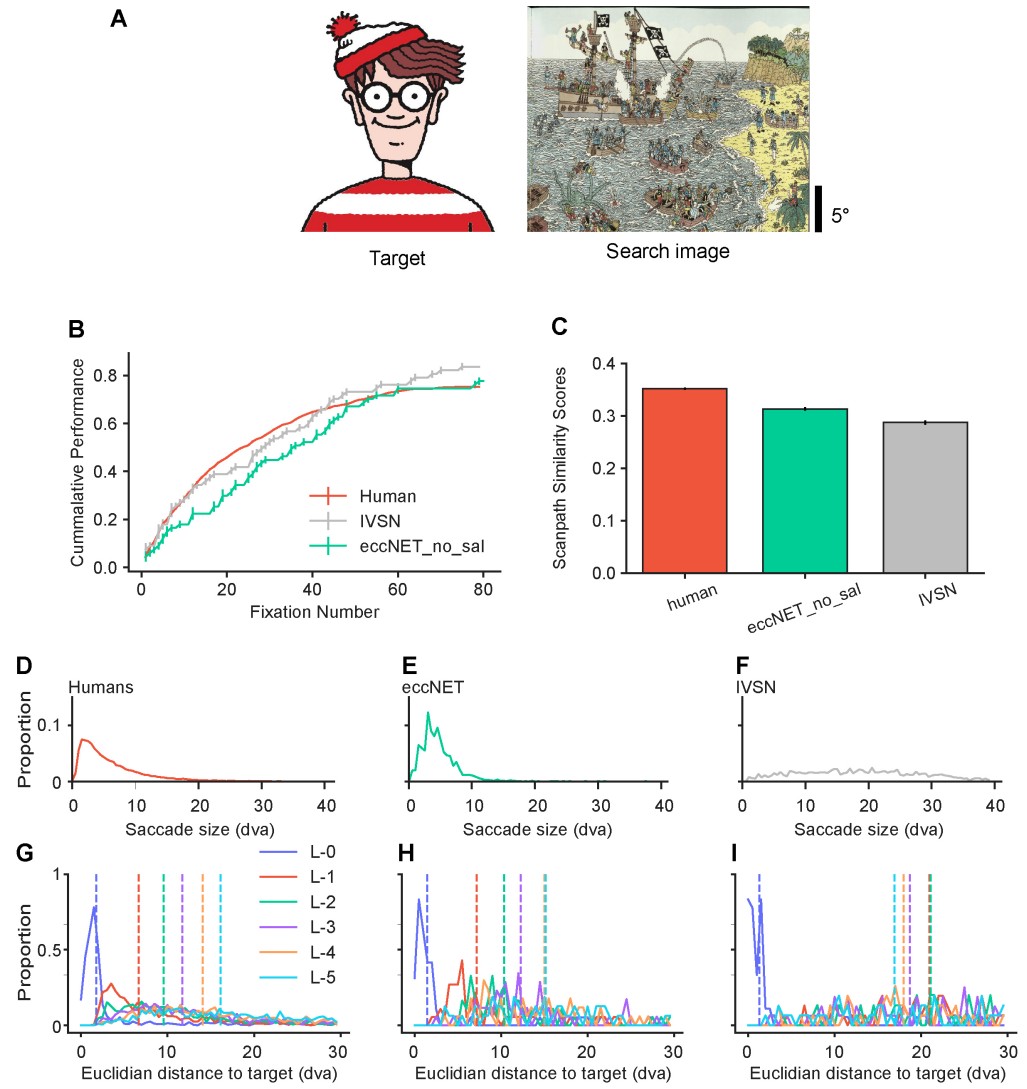

Figure S18: **The eccNET model matches previous visual search experiments with Waldo images** ([21]) The format and conventions in this figure are the same as in **Figure S16**.

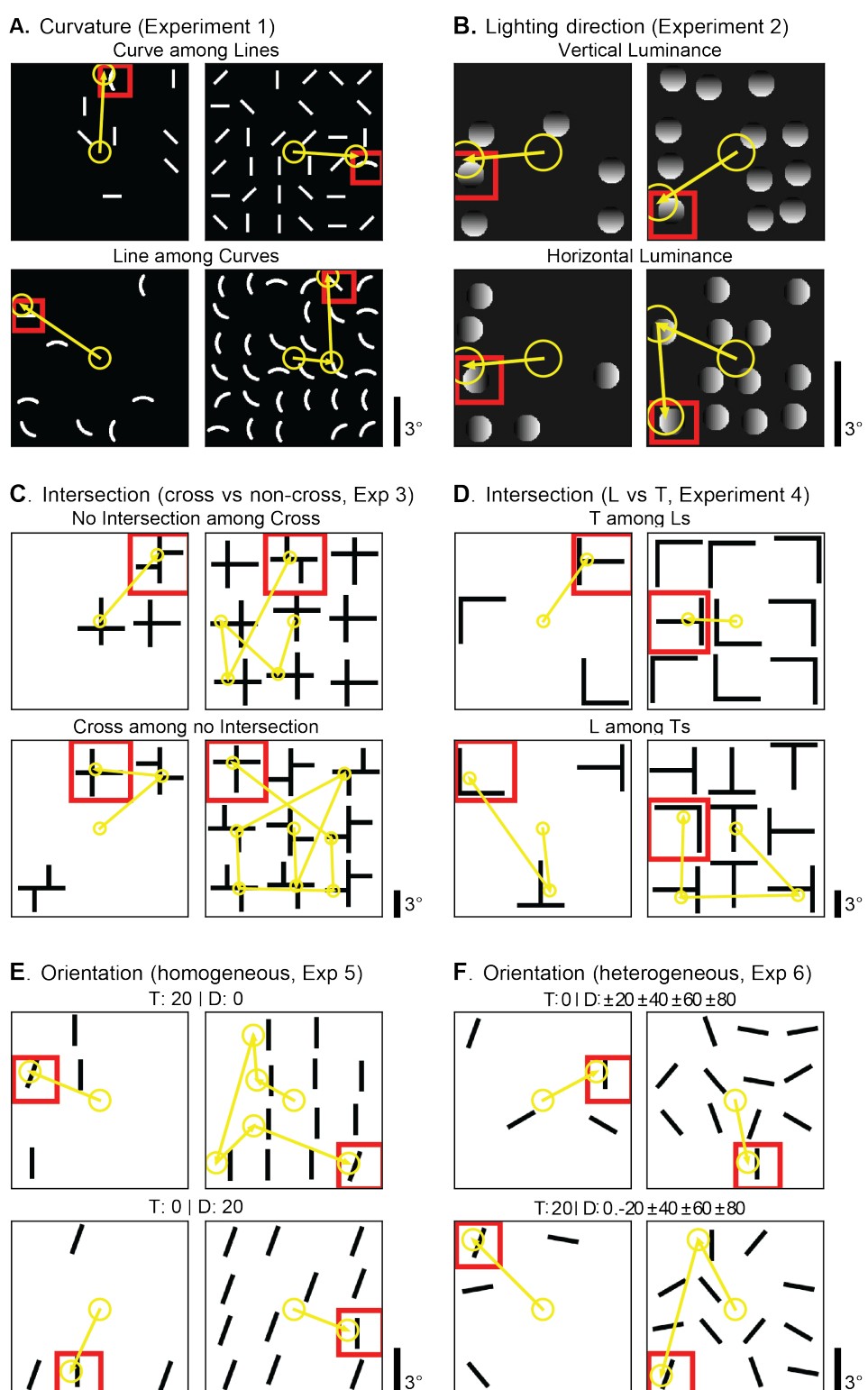

Figure S19: **Example fixation sequences predicted by the model on asymmetry search experiments.** Circles denote fixations and lines join consecutive fixations. For each experiment and condition, two examples are shown, with different numbers of objects. The red square indicates the position of the target (this red square is not present in the actual experiments and is only shown here to facilitate the interpretation of the image).

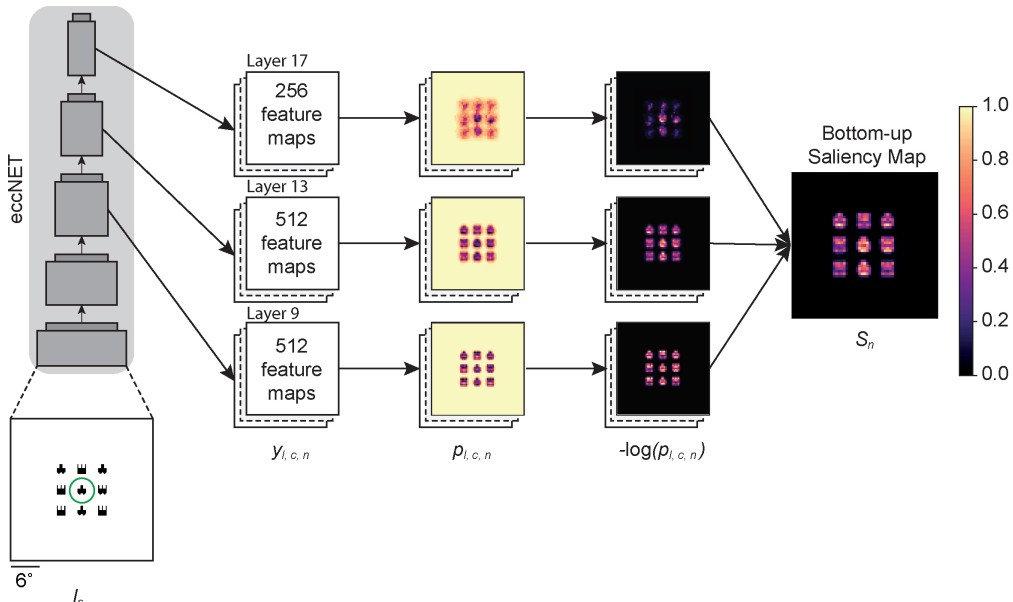

Figure S20: **Extension to the model to include bottom-up visual saliency maps (eccNET$_{bu}$).** At each fixation $n$, the saliency model extracts feature maps $(y_{l,C,n})$ at layer $l$ with $C$ channels from the visual cortex model and then estimates the probability distribution for individual channels of the feature maps $(p_{l,c,n})$. Then it calculates the self information $(A_{l,c,n} = -log(p_{l,c,n}))$, normalizes to [0,1], and adds them to compute the overall salience map $(S_n)$. See **Appendix H** section. Heatmaps show an example visualization of $p_{l,c,n}$ and $A_{l,c,n}$. See scale bars on the right for activation values on these maps.

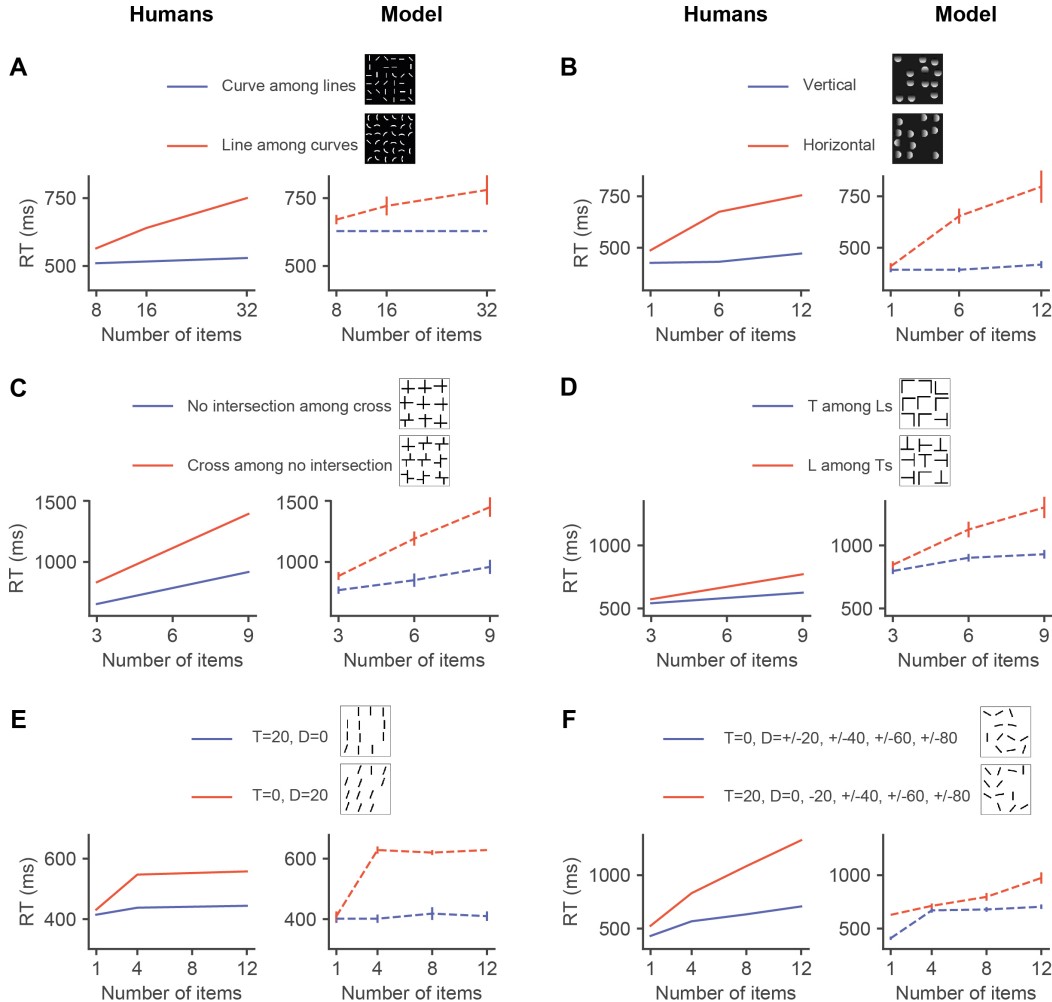

Figure S21: **eccNET_bu - Reaction time as a function of the number of objects in the display for each of the six experiments for the eccNET model with eccentricity, top-down, and bottom-up components** The figure follows the format of **Figure 4** in the main text. It should be noted that, in contrast to the main model, this version of the model uses the search data to fine tune the model.

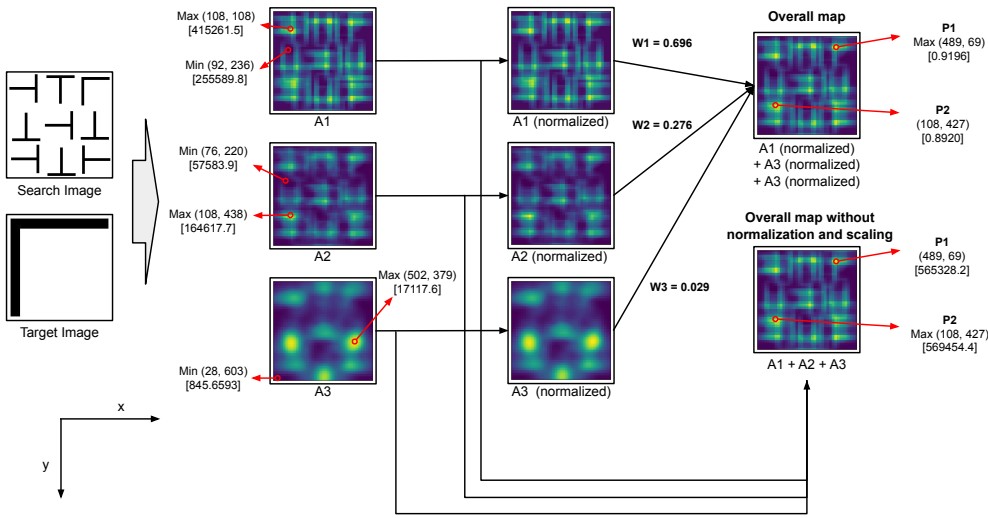

Figure S22: **Top-down attention maps. A1**, **A2**, and **A3** are the raw top-down attention maps (**Eq 2**). **Max** and **Min** are the maximum and minimum point for the corresponding maps. **A1 (normalized)**, **A2 (normalized)**, and **A3 (normalized)** are the normalized top-down maps. **Overall map** is the overall normalized and weighted top-down attention maps as per our proposed scheme (**Line 139 and 143**). **Overall map without normalization and scaling** is direct summation of **A1**, **A2**, and **A3**. Point **P1** is the max point in **Overall map** and Point **P2** is the max point in **Overall map without normalization and scaling**.

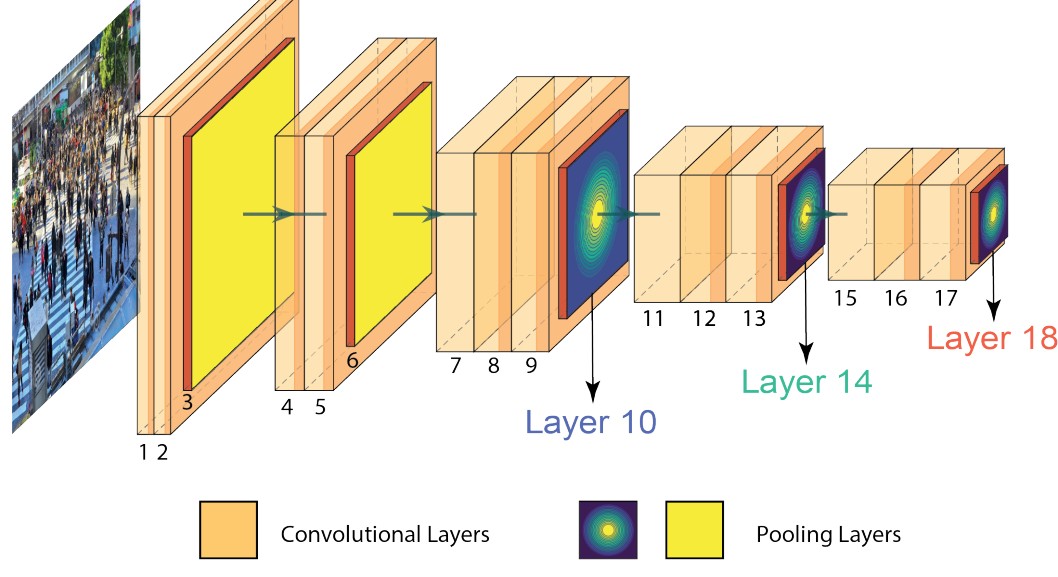

Figure S23: **eccNET feature extraction backbone showing layer numbering**
The figure shows the layer numbering for the "ventral visual cortex" feature extractor backbone of eccNET (based on VGG16) according to TensorFlow Keras [1]. This network processes both the target image and the search image, and connects to the rest of the eccNET architecture as shown in **Figure 2**.

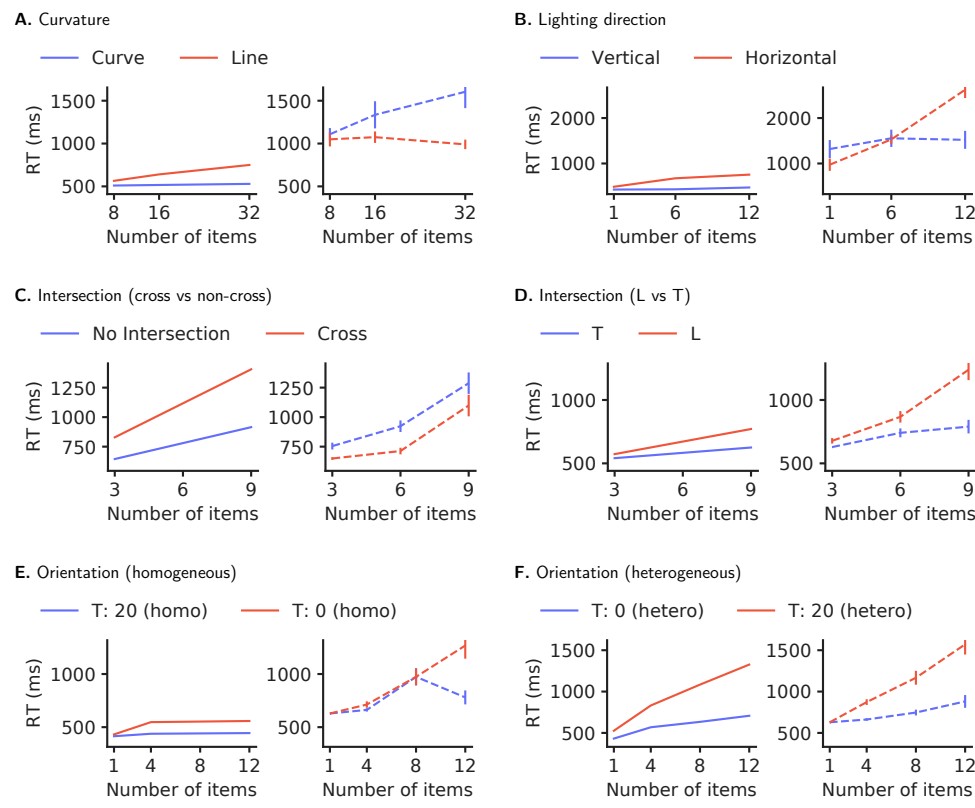

Figure S24: $ResNET152_{lastlayer}$ **- Reaction time as a function of the number of objects in the display for each of the six experiments for ResNET152** The figure follows the format of Figure 4 in the main text.