# OpenReview forum: "Visual Search Asymmetry: Deep Nets and Humans Share Similar Inherent Biases"
_NeurIPS.cc/2021/Conference — NeurIPS 2021 Poster_

### Official Review · Reviewer_F9R3 · 2021-07-14

**Rating:** 7
**Confidence:** 4

**Summary:**

The authors proposed a computational model to illuminate the mechanism underlying the search asymmetry phenomenon (finding a target A among distractors B can be easier than vice versa). Their model takes a search image and a target as inputs and produces a sequence of eye movements until the target is found. The model integrates eccentricity-dependent visual recognition with target-dependent top-down modulations. Without prior exposure to the stimuli or task-specific training, the model reproduces the polarity of search asymmetry in human vision. They also showed that when the statistics change in the dataset, the search asymmetry either disappeared or was altered.


**Limitations And Societal Impact:**

 There's no foreseeable negative societal impact of this work.

**Main Review:**

Whether the contribution is original depends on the motivation of the work and the interpretation of the results. Particularly, if the authors aim at elucidating the mechanism underlying visual search asymmetry (as stated in the abstract), then the conclusion is not very new. Visual search asymmetry has been long studied in neuroscience. Various models have been proposed to explain search asymmetry. For instance, Heinke and Backhous 2011 modeled visual search with selective attention to explain search asymmetry and showed that search asymmetry is the outcome of interactions between top-down and bottom-up processing. Various factors have been identified as sources of search asymmetry. For instance, Bruce and Tsotsos (2011) argued that stimulus familiarity determines search difficulty and may help explain search asymmetry. If the work intends to address the long-standing question of the mechanisms underlying visual search asymmetry, the authors should mention the related literature in vision sciences, including the existing theories/computational models of visual search.

By contrast, if the authors aim at showing "Deep Nets and Humans Share Similar Inherent Biases" as the title states, then the conclusion is kind of new. But the authors should mention other papers that demonstrate similar inherent bias between deep nets and humans, such as "Cognitive Psychology for Deep Neural Networks: A Shape Bias Case Study" by Ritter et al., 2017. "Bias and Generalization in Deep Generative Models: An Empirical Study" by Zhao et al. 2018 or "Number detectors spontaneously emerge in a deep neural network designed for visual object recognition" by Nasr et al. 2019.

The submission is relatively sound. The authors conducted psychophysics experiments to compare the search asymmetry in the deep nets and the ones in human subjects. They also conducted thorough ablation studies to confirm the necessity of the model components to reproduce the asymmetry. I have three major concerns. First, although the author collected the eye movement data, they only compare the reaction time and number of fixations with the model prediction. I was wondering why they did not compare the eye movement trajectory with the predicted trajectory directly. If there's a huge discrepancy between the model predicted trajectory and the human behaviors, the authors should provide some explanations.

Second, the authors claim that visual search is "the polarity of search asymmetry arises from experience with the natural environment.. is a consequence of the statistical properties of the developmental diet fed to the model." And there are only two experiments supporting that claim, which is 1) replacing the naturalistic dataset ImageNet with an MNIST and 2) rotating ImageNet images by 90 degrees. These are just some special cases of how statistics in the training set can be altered. In order to draw a strong conclusion as what the authors claimed, a much comprehensive manipulation of different statistics of the training set should be done, including contrast, spatial frequency, brightness, orientation, etc. It's entirely possible that changing some statistics won't alter the search asymmetry, contrary to what the authors found using the limited set of two experiments. In a word, I think the claim that when statistics in the training set change, the polarity of the asymmetry would be altered is a bit overgeneralizing. A minor issue related to this is that rotating 90 degrees not only changes the lighting from directions but also many other things (e.g., familiar objects such as trees and dogs would appear unfamiliar), which makes the authors' argument less convincing.

Finally, although it's nice to see that without any task-specific training, search asymmetry naturally emerges from the network, the author did not give any intuition on why adding eccentricity-dependent visual recognition with target-dependent top-down cues enables that to happen. If the author could connect that to classic theories on the human visual search to provide some intuition on why those model components give rise to search asymmetry, that could make the paper more informative and inspiring than simply showing the model works.

The submission is clearly written and well organized. If the authors could make their reasoning more solid, then the results can be important in the sense that it reveals that deep neural networks share similar inherent biases with humans when it is equipped with eccentricity-dependent visual recognition with target-dependent top-down cues. This would be useful to build AI systems that can simulate human visual search behaviors, which is extremely useful in fields like user-interface design.

------------------------------------------

After comments: I've read the authors' responses to my review as well as the ones to the other reviewers'. I think their responses address some of my concerns, especially the intuition about eccentricity and top-down modulation in asymmetry. Given these improvements, I decided to raise my score to 7.

**Time Spent Reviewing:**

10 hours

---

> ### Author Response · Authors · 2021-08-11
> **We thank the reviewer for useful feedback and interesting suggestions**
>
> 4.1. **[The goal of our work]**
>
> We thank the reviewer for pointing to these works, which we will cite and discuss in a revised version. Both of these studies are very exciting and the results are consistent with the ones discussed here. Briefly, the VS-SAIM model (Heinke and Backhous) emphasizes architectural constraints, especially the interaction of bottom-up and top-down pathways. Bruce and Tsotsos also introduce interactions between bottom-up and top-down attentional signals and additionally emphasize stimulus familiarity, which is related to the training regime noted in our study.
>
> As the reviewer also pointed out,  in the revised version, we will also survey and add related works in vision sciences on computational modelling in visual search.
>
> 4.2. **[Missing references on shared inherent biases]**
>
> We thank the reviewer again for these references. Indeed, this is the main point that we intend to emphasize and it seems that our writing was not sufficiently clear. We are certainly *not* claiming that this work constitutes the only type of shared and inherent biases between machines and humans. Nor are we arguing that this is the first demonstration of such biases. As the reviewer emphasizes in the previous question, search asymmetries have been well documented in foundational experiments in cognitive science. It is not self-evident that current deep network architectures should have the same inherent biases in this type of tasks as humans do, especially when these networks are not directly trained to search for lines, rotated Ts or intersections. In addition, there are multiple examples of dissociations between humans and machines (e.g., adversarial images, among many others). Here we use classical psychophysical findings to study these inherent architectural and training biases. It will be important to cite and discuss the studies alluded to by the reviewer and put the findings in context of other shared biases, which we will do in the revised version.
>
> 4.3. **[Comparison between human and model eye trajectory]**
>
> This is an interesting question. We focused in the paper on the keypress responses because we did not have eye-tracking data for all the experiments.
>
> For the T versus L experiment, we include a comparison of fixations predicted by ECCNet versus human fixations below, as suggested. We compare the fixations in terms of two metrics: (i) number of fixations required to find the target in each trial (the cumulative probability distribution p(n) that
> the subject or model finds the target within n fixations); and (ii) scanpath similarity score, which compares the spatiotemporal sequence of fixations (Borji and Itti,  IEEE Trans PAMI 2013, Zhang et al, Nature Communications 2018). The results  show that the model approximates, on a trial-by-trial basis, the fixations made by humans both in terms of the number or fixations and scanpath similarity score:
>
> Anonymous link for figures:[ResultsOnEyeMovementsForTvsL](https://drive.google.com/file/d/1XPVnmUI5Zb9ehL-6qdHETQCCSlRmUOTQ/view?usp=sharing)
>
> We will also include these results in the revision.
>
> The model and humans are similar but clearly not identical. These results are consistent with other  studies (which do not focus on search asymmetry) that provide a more extensive comparison of machine vs. human fixations during visual search (e.g., Zhang et al.  Nature communications 9.1 (2018): 1-15).
>
> 4.4. **[Comprehensive manipulation of different statistics of the training set]**
>
> We thank the reviewer for these interesting suggestions and comments. Because of time constraints during the rebuttal period, we could not examine the impact of the training set for all the experiments.
>
> In FigS9, we reported the results in the six experiments after the model was trained using images that were rotated 90 degrees. As the reviewer correctly pointed out, 90-degree rotation is *not* the only explanation for asymmetry. Thus, only some of the asymmetry polarities in the six experiments were reversed.
>
> The point was a proof-of-principle demonstration of how a deep convolutional neural network computational model can reveal visual search asymmetry similar to humans and thus provide a plausible mechanism for asymmetric behavior. These results suggest that humans and machines may share some inherent biases due to architecture, and development/training.
>
> There are a few interesting points indirectly alluded to in the reviewer's question. We would like to comment on them briefly here.(i) Is the mechanism for *all* forms of search asymmetry the same? We suspect that the answer is no. Thus, it is especially interesting to study the mechanisms behind other forms of asymmetry as the reviewer suggests.
> (ii) What constitutes adequate evidence in support of a model of search asymmetry? Given the assumption in (i), there may *not* be a single explanation or model for every form of asymmetry. While the accumulation of supporting evidence in the form of multiple different examples is instructive, the work here does *not* constitute a mathematical proof that the model can extrapolate to other forms of asymmetry, nor do we make such a claim. In addition to testing other forms of asymmetry we will clarify these points in the revised version.
>
> The reviewer's question also inspired us to come up with different training regimes. One of them includes using a fisheye distortion (which reduces the proportion of straight lines and increases the proportion of curves). The other one is to introduce vertical and horizontal lines, thus increasing the proportion of straight lines. We conjecture that these changes to the training set will have opposite effects on asymmetry during visual search. Unfortunately, we have not been able to finish training on these datasets during the rebuttal period, but we will also include those results in the revised version.
>
> 4.5. **[Intuition about eccentricity and top-down modulation in asymmetry]**
>
> We agree with the reviewer that it would be useful to provide a more intuitive explanation. In fact, this is one of the important criticisms of neural network models that are sometimes labeled ``black boxes'' because they may defy intuitions. The intuition here may consist of a combination of multiple phenomena rather than a single explanation. This combination of several effects is also consistent with the literature, including the work noted by this reviewer in 4.1 (Heinke and Backhous, Bruce and Tsotsos). Thus, we provide here a few points that provide some level of intuition:
>
> (i) Carrasco et al showed that there are stronger asymmetry effects in the periphery:
> Carrasco, M., McLean, T. L., Katz, S. M., and Frieder, K. S. (1998). Feature asymmetries in visual search: Effects of display duration, target eccentricity, orientation and spatial frequency. Vision Research, 38(3), 347-374.
>
> These psychophysics observations are consistent with the enhancement of asymmetry by virtue of eccentricity-dependent sampling.
>
> (ii) The combination of bottom-up and top-down signals in the model is similar to the ideas in VS-SAIM (Heinke and Backhaus 2011), who present a nice intuition in their Fig. 14 with the knowledge-based on-center-off receptive field. Although this remains to be quantitatively tested, we suspect that units in the current model are also likely to show competition between dominance by bottom-up search image features or top-down guidance (referred to as template matching by Heinke and Blackhaus).
>
> (iii) Bruce and Tsotsos (2011) emphasize familiarity and the intrinsic biases and structure of feature representations in visual cortex. The ablation studies (Fig. 5) and the modifications of the training regime (Fig. 6) are consistent with these ideas.
>
> We will expand on the discussion to provide a better account of possible intuitions behind the model's inner workings.
>
> 4.6. **[Summary]**
>
> We thank the reviewer for summarizing our work better than we did.
> As the reviewer suggested, we will make the reasoning (intuition about eccentricity and top-down modulation in asymmetry) stronger (also see the response to the previous question).
> In fact, we would like to "borrow" parts of the summary by the reviewer verbatim in our revised version. There are ongoing debates about whether building models that can match human behavior is a useful approach in AI or not and the reviewer articulated these ideas better than we did!

---

### Official Review · Reviewer_CJM4 · 2021-07-14

**Rating:** 7
**Confidence:** 3

**Summary:**

This paper proposed an image-computable model to examine potential mechanisms underlying asymmetries in visual search. The proposed method can capture similar behaviour as the human user, and the proposed architecture is novel. However, I am not fully convinced why we need to investigate the asymmetries with the neural networks model, and I am not sure how much of the training data instead of the proposed method can affect the results.

**Limitations And Societal Impact:**

The author provides a detailed discussion on limitations. I do not think there is nay negative societal impact of this work.

**Main Review:**

**Paper strength**
- The paper is well-written. The author provides sufficient details of the implementation of the proposed method, and a good illustration of the methods as well as the experiment results.

- The proposed method has a certain novelty. The eccentricity-dependent pooling is meaningful for such an attention-based model, and the author extracts the feature according to the number of the layer. The author also proposes a top-down attention modulation. Both two components are evaluated with the ablation study to show their effectiveness (Sec. 4.3).

- The experiment (Fig. 4) shows the proposed method successfully capture the visual search asymmetry of the human user. It is an interesting finding that the pre-trained model on ImageNet has such ability. In comparison with other baselines in Fig. 5, the proposed method shows it achieve leading performance and similar to the performance of the human user.

- The author provides interesting discussions on where the asymmetry comes from in Sec. 4.4. The author thinks the asymmetry comes from the lighting direction within the natural images.

**Paper weakness**
- It is not clear to me why we want the trained model to have similar behaviour as the human. Especially, why we want the model to capture such asymmetry as the human being? I understand that the current machine and human have their own advantage respectively, and it is not necessary that the machine must learn from the human.

- One experiment shows such asymmetry also depends on the training data. The model trained on the MNIST dataset does not have an asymmetry effect. It shows that such asymmetry actually depends on the data instead of the method/architecture. If we want to investigate the asymmetry, we should perform the analysis on the data distributions.

- It will be very interesting if the author found the asymmetry is a general effect for neural network models. However, it is not the case as shown in Fig. 5. The asymmetry can be captured well with the proposed method, not other neural networks models. In this sense, it is not clear to me why the author wants to build such a model.

-The assumption that the asymmetry comes from the lighting condition is not convincing. It might be true for the outdoor environments but not for the indoor setting. It will be better if the author could provide any static numbers to support their argument. Such an assumption also cannot explain other shapes such as the curve.

- I would not agree that the classification result of 66.7% is similar to 72% (line 267).

**Time Spent Reviewing:**

3

---

> ### Author Response · Authors · 2021-08-11
> **We thank the reviewer for useful feedback and interesting suggestions**
>
>
> 3.1. **[Motivation of building human-like computational models]**
>
> It is important to emphasize that we are *not training* any parameter in the model to fit the human behavioral data in the results shown in the paper. And yet, it is interesting to observe that deep neural networks, pre-trained on ImageNet for object recognition task, *without* any human data fitting, *still* show similar asymmetry patterns as humans. These results suggest potential shared inherent biases in the architecture and training between humans and machines. We agree with the reviewer that whether machines need to learn from humans to become smarter or not, constitutes a matter of extensive debate in AI. We sympathize with the reviewer's viewpoint. From an applications standpoint, we want algorithms that work irrespective of whether they are better, worse, similar or different to humans. Machines can solve problems in ways that are completely different than humans. However, we argue, with many others, that building models that can capture human behavior can be quite instructive in many ways (e.g., Hassabis et al, "Neuroscience-inspired Artificial Intelligence", Neuron 2017; Lake et al, "Building machines that think and learn like people", Behavioral and Brain Sciences 2016, Kreiman, "Biological and Computer Vision" Cambridge University Press 2021; Poggio and Anselmi, "Visual Cortex and Deep Networks", MIT Press 2016): (i) to better understand human cognition, and potentially alleviate brain disorders, (ii) because humans still outperform machines in the majority of cognitive tasks despite exciting progress in the field; (iii) because biological brains are the product of millions of years of evolution and have found solutions that are often more robust, efficient, and work in the real world. We argue, with others, that we can learn from human brain computations to build better algorithms.
>
> As an analogy, adversarial images illustrate how human perception can deviate from machine vision. Similarly, search asymmetry constitutes another curious observation of how humans use top-down guidance in an everyday task, i.e., searching for an object. The results here provide a proof-of-principle demonstration that neural network models *can* capture this type of behavior *without any training* on human behavioral data.
>
> 3.2. **[Characteristics of training data]**
>
> We argue that visual search asymmetry behavior is a consequence of **both** architecture design as well as the training data. In the ablation study (Fig5B), we removed eccentricity-dependence as well as top-down modulation across multiple layers. Both ablated models show lower average asymmetry index compared with our full model. This suggests that in addition to feature biases in the training set, the architecture design is also important to show asymmetric visual search.
>
> We agree with the reviewer that the analysis on data distributions is interesting. Given the complex nature of existing large image datasets, digging out exactly what the inherent biases are is often a challenging task. Thus, we emphasize that visual search asymmetry experiments might be one of the useful avenues to explore these biases.
>
> 3.3. **[No asymmetry in other neural networks]**
>
> A positive asymmetry index indicates that a model shows visual search asymmetry. In Fig 5B, EccNet as well as all the ablated models show visual search asymmetry. The baselines that do not show search asymmetry are not based on deep networks (such as pixelMatch and gbvs). We will emphasize this point in the final version. In addition, here we use VGG16 as the backbone of EccNet. Similar feedforward object recogniton networks (such as ResNet) could also replace VGG16 as the backbone here to show such search asymmetry behavior. Though ResNet does not approximate human behaviors as well as EccNet, it still shows similar asymmetry behavior (positive asymmetry search index) in four out of the six experiments.
>
> Anonymous link for figures[AsymmetryResNetAsFigure4](https://drive.google.com/file/d/1CIBpcO6x0rJOP7SFscOxzuwcFqpQtRKX/view?usp=sharing)
>
> We will add and include these results in the revision.
>
> Thus, all the neural network models that we tested show visual search asymmetry. Of course, this does not demonstrate that visual search asymmetry is a general effect for *any* possible neural network. The results also show that the amount of search asymmetry depends on the network architecture and also on the training set. Some models (EccNet) are more similar to human behavior.
>
> 3.4. **[Lighting condition in indoor setting]**
>
> To further evaluate this question, we will train the network using the indoor images in Places dataset (places2.csail.mit.edu). We conjecture that even in the indoor settings, most of the light sources are installed on the ceiling where lights also come from above (e.g. ceiling light). Of course, this is a conjecture that needs to be quantified. We will add these analyses in the revised version.
>
> 3.5. **[Asymmetry in shapes such as curves]**
>
> We agree with the reviewer. We are *not* claiming that lighting direction explains *all* asymmetry conditions. We provide this as an example to illustrate how the statistical properties of the training diet can lead to asymmetry. As the reviewer rightly points out, there are many other statistical regularities in the data in addition to lighting direction. For example, natural images tend to have a disproportionately high preponderance of cartesian directions (horizontal and vertical) compared to other directions, which may underlie the asymmetries based on orientation. Similarly, there is also a relatively high proportion of straight edges compared to curvatures, which may underlie the asymmetries based on lines versus curves.
>
> As the reviewer suggested, we will conduct two additional data augmentation experiments: (i) we will apply a fisheye transformation on all training images of ImageNet (to reduce straight lines); (ii) we will add rectangular grids on all training images of ImageNet (to increase the proportion of straight lines in the training set exposed to the model). Due to time constraints, we have not yet got the results but we will include all these results in the revised version.
>
> 3.6. **[Line267: 66.7% is similar to 72%]**
>
> We agree with the reviewer. First, in the revised version, we will add the standard deviation to both numbers to provide a better sense of their difference and variation. Second, we will remove the statement indicating that they are similar. The main point that we were trying to make in line 267 is that the changes introduced in the model do not completely disrupt object recognition performance. We are not optimizing for object recognition performance and we are *not* claiming that the proposed model performs better than VGG16 (let alone state-of-the-art models) in object recognition. Instead, we argue that features learned for object recognition can transfer to a different task, i.e., visual search. Furthermore, and quite remarkably in our view, those features generalize to capture essential aspects of human experiments with simple shapes, experiments which have been foundational in a different field, cognitive science.

---

> > ### Comment · Reviewer_CJM4 · 2021-08-27
> > **The author has addressed my concerns**
> >
> > I thank the author for their professional response.
> > I agree with the author that capturing human behaviour can be quite instructive, which is meaningful for future research. I thank the author for their detailed explanation and for pointing out related works.
> >
> > I also agree with the author that there are many other statistical regularities in the data in addition to lighting direction. I hope the author could add more discussion on the potential causes of the asymmetry.
> >
> > For the additional results on AsymmetryResNetAsFigure4, it requires permission to check.
> >
> > The author addressed all of my concerns about this paper and promised extensive experiments that were more than I expected. I would like to raise my rating accordingly.

---

> > > ### Author Response · Authors · 2021-08-28
> > > **Thank you**
> > >
> > > We thank the reviewer for the feedback.
> > >
> > > Yes, as we promised in the rebuttal, we will add more discussions about other causes of the search asymmetry in the final version of our paper.
> > >
> > > We apologize for the incorrect permission to view the figure. We have now corrected the permission. Please see the anonymous link for figures [AsymmetryResNetAsFigure4](https://drive.google.com/file/d/1CIBpcO6x0rJOP7SFscOxzuwcFqpQtRKX/view?usp=sharing)

---

### Official Review · Reviewer_CMyL · 2021-07-14

**Rating:** 5
**Confidence:** 5

**Summary:**

In this manuscript, the authors present a visual search model, which predicts a fixation order over visual search displays. The model is based on VGG-16 replacing the max-pooling layers with average pooling with growing pooling area towards the periphery. The priority for fixation is then computed as a combination of target-image similarity at three different feature maps in the network. The primary result reported is that model qualitatively reproduces 5 out of 6 visual search asymmetries.

**Limitations And Societal Impact:**

The authors do not comment on their limitations and societal impact. Besides the sampling biases of the used databases I don’t think there are important societal impacts associated with the content of this paper though.

**Main Review:**

Overall, the observation that visual search asymmetry might be caused by peripheral pooling is interesting and the model the authors use seems reasonable. I have a relatively long list of questions though and in quantitative terms the fit of the model is bad. Thus, I do not find the model and manuscript convincing yet.

My first questions are on Figure 3: Why didn’t the authors choose the pooling areas such that the eccentricity dependence follows the ones of the visual areas they compare to? Also, are there any reasons why Layer 10 should be associated with V1 etc?
Also, the illustration in A appears to be wrong as the center is not blurred at all, while the formulas give a 2 pixel radius pooling for the fovea at each location, which should correspond to noticeable blur in the center.

I find the pooling operation across attention maps strange. Why are the 3 A maps scaled as they are? The overall scaling amounts to max(A) / (max(A) - min(A)), i.e. a stronger weighting the larger the minimal value is in proportion to the maximum. How does this make any sense?

Also I find the nomenclature of calling the maps which measure target-image similarity at the different layers (A_{i \rightarrow j}) very odd as it strongly implies a modulation of the processing within the network producing the activations. If I understand the manuscript and the illustration in Figure 2 right, this does not happen though, but the convolution is only applied to create the readout maps.

On the pixelMatching model: As all search images with the target contain an exact copy of the target: Why doesn’t the pixel matching model find all targets within the first fixation it chooses?

As the model produces whole scan paths: Have the authors considered looking at more detailed aspects of them rather than just the overall duration?


----

After comments:
I have just slightly increased my evaluation of this manuscript as the authors have addressed some of my concerns. Especially they now do some comparisons on the scan-paths and show some better quantitative fits to the reaction time data.
Also reading some of the other reviews highlighted some of the advantages of this manuscript.

Unfortunately, the authors' answers didn't remove my concerns about the model though. Some choices still appear strange/unusual or not exactly representative for the described mechanisms to me. For me this casts doubt on any of the interpretations, which of course rest on the assumption that the model is a good representation of the theoretical ideas.
It is entirely possible, that my concerns can be addressed and this manuscript ends up being correct, but I honestly see some probability that the results do depend on some oddity of the model and are not interpretable as they are interpreted here.

Thus, I won't improve my rating further here, but suggest some clarification and maybe some controls as suggested by other reviewers may help for the future.

To be concrete:
- on 2.2 / 2.3.: I still don't understand why the pooling regions were not exactly matched to the monkey data. Scaling pixels and the slope should allow any linear function and thus a perfect fit between the two figures. Little of the conclusions in this manuscript rests on the question which area corresponds to which layer, but not making them fit where you clearly could is strange.
- on 2.5: I did understand the scaling of the attention maps. My criticism is that the described scaling does not appear sensible to me. As you weigh maps by their maxima after scaling them by their range the difference in scale is restored and each map is effectively scaled by how much of its range is above 0, which seems highly unusual to me.
- on 2.6: I originally formulated my criticism here aimed at changing the formulation, thinking that the authors "obviously" cannot mean to interpret this interaction as top-down modulation. It seems that they do mean to interpret this as an implementation of top down modulation. I strongly disagree with this interpretation.
The two interacting layers are a feature map and a blurred version of the same feature map (or RELU transformed, I am not perfectly sure due the confusing layer numbering) . Thus, the operation is a template match with a slightly blurred representation of the target (slightly, because the target image is processed at fixation if I am not mistaken). Template matching is a typical strategy used in search models, but has no connection to top-down modulation for me. If there is any top-down connection it is from working memory to the feature map not between layers of processing.

Note on the last point: If the two layers were not using closely related feature maps, convolving the feature map with a target representation in the higher layer would not make any sense and would in many cases not be possible due to a mismatch in the number of features.

By the way: Something appears to be off about the counting of layers the authors use. They write that layer 10 corresponds to the 6th convolutional layer, but I checked and VGGs use RELUs after each convolution, such that it could be at most the 5th and in fact guessing based on the figure and a look at the pytorch hub VGG could be either the second max-pooling layer, which appears after 4 convolutions or the third, which happens after 7 convolutions though and has a layer index of 16 in pytorch.


**Time Spent Reviewing:**

11

---

> ### Author Response · Authors · 2021-08-11
> **We thank the reviewer for useful feedback and interesting suggestions**
>
> 2.1. **[Quantitative fits]**
>
> We compared eccNet and human performance in terms of the reaction time (Fig4) and Asymmetry Index (Fig5). Quantitative results using both metrics indicate that our model outperforms all baselines and share high similarity with humans in asymmetric search behaviors. However, the reviewer rightly points out (as we emphasize in the paper) that the model often does not make precise predictions of absolute reaction times.
>
> It is important to emphasize that we are *not fitting* any parameter in the model to the behavioral data in the results shown in the paper. Given the lack of parameter tuning, and the multiple differences between humans and machines (e.g., how humans are "trained", target-absent trials only for humans, motor cost for humans, target localization versus detection), one may not expect precise fitting of reaction times.
> What we find remarkable is that even without such parameter tuning, it is possible to capture fundamental properties of human behavior. We consider the results to demonstrate as a proof-of-principle, that neural network models can show the type of asymmetric properties that are evident for humans.
>
> In addition, if we allow us to fine tune the model while using cross-validation to avoid overfitting, as is customarily done in most computer vision applications, it is possible to obtain tighter quantitative fits to the reaction times.
>
> See anonymous link: [ReactionTimeModelFit](https://drive.google.com/file/d/1BAZOnknJdOHKxROrzWs2siDgmMAg6F6i/view?usp=sharing).
> We will add this figure and discuss the points above in the revision.
>
> 2.2. **[Pooling areas]**
>
> First, we map receptive field sizes measured in degrees of visual angle to pixels; this process is explained in Appendix D (page 3 of the Supplement) and Table S1.
> Second, we describe the eccentricity-dependent pooling mechanisms in the model on pages 4-5 (Equation 1).
> Both of these steps are meant to approximate the eccentricity-dependent sampling in different areas of the ventral visual cortex.
> Next, we measure receptive field sizes for units in the model at different eccentricities (Figure 3B, left). The model shows a coarse approximation to the empirical observations (compare Figure 3B, left versus right). For example, layer 10 in the model is approximately similar to V1 (blue), etc. The model is deliberately simplistic, with two parameters (slope of eccentricity versus receptive field sizes $\gamma$, scaling factor converting degrees of visual angle to pixels $\eta$), and there is certainly ample room to build better approximations of the receptive field sizes. It should also be noted that the data in Figure 3B, right panel, comes from monkeys whereas all the behavioral data comes from humans. To the best of our knowledge, there are no neurophysiological measurements of receptive field sizes at different eccentricities from human ventral visual cortex. Some investigators have used field potential measurements (e.g., Agam et al Current Biology 2010), but those experiments are not as complete as the monkey neurophysiological measurements.
>
> 2.3. **[Layer 10 associates with V1]**
>
> First, it is worth noting that we are using layer numbers based on the convention in Pytorch where even the activation function (ReLU) is a separate layer. Thus, layer 10 corresponds to the 6th convolutional layer in the VGG16 architecture.
> Still, the reviewer rightly ponders why V1 is mapped onto this 6th processing stage. Other investigators have assumed that V1 maps onto layer 1, or the first convolutional layer (this is often not stated explicitly but it is often implicitly or indirectly assumed; see for example the illustration of orientation tuning in layer 1 in Kryzhevski et al, NeurIPS 2012).
>
> Eccentricity-dependent sampling begins at the retina, and is evident at all processing steps, including the lateral geniculate nucleus, V1, etc. Mapping V1 onto the 6th processing stage may actually be more biologically plausible than assuming a single transformation between pixels and V1 (though one could argue that this transformation includes all the extensive processing that takes place between photoreceptors and V1).
>
> The main goal of the current study is to evaluate the asymmetry in visual search behavior. However, we agree with the reviewer that more work needs to be done to better understand how different layers in computational models map onto different processing stages along the visual system. Furthermore, it is important to make a commitment about these mappings. While this is beyond the scope of the current work, this is fertile territory for further work.
>
> 2.4. **[Blur in the center]**
>
> We think that this is due to the resolution of the figure in the paper. The example image in Fig3A is 512 x 512 pixels, equivalent to about 17 x 17 degrees of visual angle.
> Given the current layout of the paper, it is difficult to see noticeable blur around the center with 2 pixel radius pooling. We are providing an enlarged version of the figure that can be zoomed in to assess the small amount of blurring in the center.
>
> See anonymous link [ZoomedInVersionFig3A](https://drive.google.com/file/d/1dRUuR-4Uq8NVazub9pXKx3qeyXnZ1NBh/view?usp=sharing)

---

> ### Author Response · Authors · 2021-08-11
> **We thank the reviewer for useful feedback and interesting suggestions - Part 2**
>
> **This is the second part of our response**
>
> 2.5. **[Scaling of 3 attention maps]**
>
> We have three top-down modulation attention maps from three layers. They have different scales. As noted in the ablation studies in Figure 5B, the multi-scale attentional modulation provides a better description of the human behavioral data: a single map (e.g., layer 18 modulating layer 17) leads to a lower average asymmetry index. Given that we have three attention maps at different scales, the model needs to combine them somehow to direct the next fixation. The reviewer rightly points out that there could be several ways of combining those individual attention maps. We opted for what we consider to be a rather simple and plausible way of combining them by using a weighted linear combination of normalized attention maps. The normalization is important because the raw activation values vary substantially across different layers; relevant locations are those that are salient according to the similarity to the target at each scale. Overall, the scaling depends upon the max value of individual attention maps. Inserting the weighting values (line 143) into the equation combining the three maps (line 139), the overall scaling factor for attention map $A_i$ is:
>
> $\frac{max(A_i)}{(\sum_i max(A_i))*(max(A_i)-min(A_i))}$.
>
> We will add this equation and better explain the motivation for creating separate preliminary attention maps at different scales, for normalizing them, and for combining them with a weighted average. We have not examined other ways in which the preliminary attention maps can be combined. The reviewer is right to highlight this point, which opens the doors to interesting possibilities. For example, it may be that different tasks could rely more on some scales compared to others.
>
> 2.6. **[Attention map nonmenclature]**
>
> We are afraid that we do not fully understand the question here. We are not exactly sure what the reviewer refers to as ``readout'' maps. The question emphasizes that we did not explain the generation of the attention maps in a clear manner. Briefly, we have a set of activations/features that represent the sought target. Say that the model is searching for a curved shape. The target is represented in different layers (which can be thought of as information about features of the sought curved shape at multiple scales, in multiple areas along visual cortex, lines 130-132). Next, the model is presented with a search image. At each fixation, the model has a set of activation/features that represent the search image (again, at multiple scales, here dependent on the fixation location, lines 132-133). Next, the model creates three preliminary attention maps by top-down modulation between consecutive layers at three different scales (Equation 2). For example, target activations/features from layer 18 provide top-down modulation onto search images activations/features at layer 17. Of note, these maps are recomputed at each fixation. Next, these three preliminary attention maps are combined into a single attention map. This overall attention map is then used to select the next fixation location (lines 137-143). Our explanation in the paper and here is rather succinct. We will provide a more detailed step-by-step explanation in the revised version and we will accompany this with a figure that describes all the intermediate steps as briefly outlined here. If this is still confusing or does not address the reviewer's question, we would appreciate further clarification as to which steps are not well explained.
>
> Given these steps, we maintain that the nomenclature is adequate. The model is using top-down modulation from layer l+1 to layer l. The additional sub-index n is relevant because these maps depend on the fixation location (n is the fixation number).
>
> 2.7. **[pixelMatching model]**
>
> The reviewer's intuition is quite right. The pixelMatching model does find the target within the first fixation in the majority of cases (see Figure S5).
>
> The reason that it is not 100 \% of the cases the first fixation is that
> the matching procedure also depends on the stride of the sliding window (i.e., how many pixels to jump at every step). Still, the reviewer had the correct intuition. Humans do not seem to use this type of template matching strategy and this model does not adequately capture human performance.
>
> 2.8. **[Detailed aspects of scanpaths in T vs L experiment]**
>
> This is also an interesting question. We focused in the paper on the keypress responses because we did not have eye-tracking data for all the experiments.
>
> For the T versus L experiment, we include a comparison of fixations predicted by ECCNet versus human fixations below, as suggested. We compare the fixations in terms of two metrics: (i) number of fixations required to find the target in each trial (the cumulative probability distribution p(n) that the subject or model finds the target within n fixations); and (ii) scanpath similarity score, which compares the spatiotemporal sequence of fixations (Borji and Itti,  IEEE Trans PAMI 2013, Zhang et al, Nature Communications 2018). The results  show that the model approximates, on a trial-by-trial basis, the fixations made by humans both in terms of the number or fixations and scanpath:
>
> Anonymous link for figures: [ResultsOnEyeMovementsForTvsL](https://drive.google.com/file/d/1XPVnmUI5Zb9ehL-6qdHETQCCSlRmUOTQ/view?usp=sharing)
>
> We will also include these results in the revision.
>
> The model and humans are similar but clearly not identical. These results are consistent with other  studies (which do not focus on search asymmetry) that provide a more extensive comparison of machine vs. human fixations during visual search (e.g., Zhang et al.  Nature communications 9.1 (2018): 1-15).

---

> ### Author Response · Authors · 2021-09-24
> **We thank the reviewer again for useful feedback and interesting suggestions**
>
> **[On 2.2 / 2.3.: I still don't understand why the pooling regions were not exactly matched to the monkey data. Scaling pixels and the slope should allow any linear function and thus a perfect fit between the two figures. Little of the conclusions in this manuscript rests on the question which area corresponds to which layer, but not making them fit where you clearly could is strange.]**
>
>
>
> 1. There are various constraints and computational limits for making a perfect fit:
>     1. Images are represented as “quantised pixel units”, i.e.,  we have limited pixel sizes to use.
>     2. Scaling the input image size can be done to map some fractional window size of “0.5x0.5” equivalent to some integral window size of “2x2” or “3x3”. But this comes at the cost of using a large size of the input image. There’s a memory limitation on the GPU front on how large the images we can use are.
>     3. In principle, we could use interpolation between the neighbouring pixels while applying the pooling operation but we have not tried this.
> 2. It is worth pointing out that the curves shown for macaques, as reproduced in Fig. 2B right, constitute *average* measurements. There is considerable variation in the receptive field sizes, even at a fixed eccentricity and fixed visual area. As one example of many, consider the variation in Figure 9 in Kobatake and Tanaka (Journal of Neurophysiology 1994).
> 3. It is also worth pointing out that there are extensive measurements of receptive field sizes of individual neurons in macaque monkeys (and also cats and rodents), but there is essentially no such measurement for humans. There exist field potential measurements of receptive fields in humans (e.g. Yoshor et al Cerebral Cortex 2007 for early visual areas and Agam et al Current Biology 2010) for higher visual areas). Thus, even if we strived to make a better fit to the average macaque data, it is not very clear that this would help us better understand the behavioural measurements in this study which were conducted in humans.
>
>
> **[On 2.5: I did understand the scaling of the attention maps. My criticism is that the described scaling does not appear sensible to me. As you weigh maps by their maxima after scaling them by their range the difference in scale is restored and each map is effectively scaled by how much of its range is above 0, which seems highly unusual to me.]**
>
> Each map is first normalized and then is assigned with an individual weight. Our design methodologies express that each of the top-down attention maps corresponds to top-down modulation from different sets of unique features.
>
> Since these features are unique to the target image type, the weight given to them must not necessarily be equal and should depend on the demands of the given task.
>
> One way to get weights is by doing a parameter fitting. But to avoid parameter fitting specific to the given visual search experiment, we considered using the max value predicted by respective top-down activation maps to decide the weights.
>
> **Why does max activation make sense?**
>
> Since top-down activation maps are produced due to modulation from the target image to the search image feature, a higher activation value will mean a high similarity between the target and search image at the corresponding feature level. Thus, giving a higher weight to the attention map produced using that feature layer makes sense. It ensures that the model will provide higher importance to the attention maps at those feature layers which are more prominent in the target image.
>
> **An example of normalization and weighting.**
>
> To illustrate the point that it’s not necessary that the map will get restored we are adding an example case with all the calculation and attention maps generated at intermediate steps leading to different results at the end as compared to an overall map which is estimated by direct addition of the three top-down attention maps without any normalization and weighting.
>
> Anonymous link for figure and calculations: [ [TopDownAttentionMapCalculation](https://drive.google.com/file/d/1o9y4U4R0vzw9GaaoDafBKhW6LP7NTghV/view?usp=sharing) ]
>
> **[On 2.6: I originally formulated my criticism here aimed at changing the formulation, thinking that the authors "obviously" cannot mean to interpret this interaction as top-down modulation. It seems that they do mean to interpret this as an implementation of top down modulation. I strongly disagree with this interpretation. The two interacting layers are a feature map and a blurred version of the same feature map (or RELU transformed, I am not perfectly sure due the confusing layer numbering) . Thus, the operation is a template match with a slightly blurred representation of the target (slightly, because the target image is processed at fixation if I am not mistaken). Template matching is a typical strategy used in search models, but has no connection to top-down modulation for me. If there is any top-down connection it is from working memory to the feature map not between layers of processing. ]**
>
> First, the reviewer is correct that the target image is processed at fixation.
>
> We suspect that the reviewer correctly interpreted what we are doing and the discrepancy might be an issue of nomenclature. We use the term “top-down” to refer to the fact that the model stores information about the target and uses this task-dependent information to modulate the responses by convolving features from layer l+1 for the target with those features from layer l in the search image.
>
> Perhaps the reviewer is less interested in the biological implementation, but we think that it is highly likely that this process happens via top-down modulation in the brain. What would be the alternative? The subject is shown the target, a given area (say V4) extracts features and stores those features, and then upon presentation of the search image, the same V4 neurons need to compare their memory of the target with the new incoming information. Indeed, there is evidence of working memory representation along the ventral visual cortex (e.g. for area V4, Chelazzi et al Cerebral Cortex 2001, among many others). However, this process seems unlikely to us because the same V4 neuron would have to store previous information and compare that with the new input. An alternative would be to have two sets of V4 neurons, one set involved in working memory and storing features and another set that processes incoming information. While this option seems easier to implement, we do not know of any evidence of a separation between neurons involved in working memory exclusively and other neurons involved in processing incoming inputs exclusively. We think that it is more likely that the target information is stored in a different brain area. We postulate the pre-frontal cortex as a likely candidate, e.g. Miller, Working memory 2.0, Neuron 2018, as one of many examples showing working memory representation in the pre-frontal cortex. The information from the pre-frontal cortex then propagates back in a top-down fashion, modulating the responses along the ventral visual cortex.
>
> We can still call this process “template matching”. But we would like to emphasize that: (1) it comes from the top-level representation of the target image and propagates backwards, (2) it is unlikely to be pixel-level matching (see for example Zhang et al Nature Communications 2018 for invariant visual search).
>
> **[Mismatch in the number of features during top-down modulation]**
>
> As the reviewer correctly pointed out, the choice of the units that receive top-down modulation in layer l is not randomly chosen. We selected the units in layer l such that the units in layer l+1 correspond to be the pooling operation of those units in layer l. Thus, the number of features at layer l+1 are still consistent with those in layer l. Thus, there is no mismatch in the number of features.
>
> **[Layer numbering]**
>
> First, we would like to clarify that all the code was written using Keras and TensorFlow, not PyTorch (Page 4, Section G in the supplementary material).
>
> Second, we have provided all the code for replication as supplementary material during the paper submission. Specifically, the directory: **/vs_model/ecc_net.py **and function name **load_eccNET()** is where one can verify the exact layer numbering.

---

### Official Review · Reviewer_h9eJ · 2021-07-17

**Rating:** 8
**Confidence:** 3

**Summary:**

This paper proposes a computational model (eccNET) that produces visual search asymmetry patterns similar to those seen in humans. The model is pretrained on ImageNet and integrates eccentricity and top-down cues to produce a series of fixations until the target is found. Since a rotation of the training set changes the pattern of results, the authors conclude that visual search asymmetry may be a consequence of natural image statistics.

**Limitations And Societal Impact:**

The authors have addressed the limitations of the work.

**Main Review:**

The goals of this paper appear to be twofold: first, to understand whether visual search asymmetry of the type seen in psychophysics experiments emerges in neural networks not specifically trained in visual search; and second, to understand the mechanisms through which this happens by manipulating the model architecture and training set.

The paper takes a thorough approach both in characterizing human behavior (using data from six experiments) and machine behavior (four baselines, as well as model ablations to assess the role of model features in matching human behavior). I found that many questions that arose while reading the paper had been addressed by the end. The paper is well-written and well-structured.

The authors used data from six psychophysical experiments with different visual search tasks showing asymmetry in humans. Since these experiments relied on target-present/target-absent keypress responses, the authors collected an additional eyetracking dataset to aid in linking fixation data from eccNet to RT data in humans. Limitations of this approach are discussed in the paper; my only comment is that it would be interesting to see how the fixation data compares to eccNet in this one experiment (despite all the expected differences that the authors discuss).

In terms of pinpointing hypotheses for why this asymmetry emerges in eccNet, the paper seems to hint at a conclusion. First, it appears that biologically plausible model features are crucial (eccentricity and top-down feedback at several layers have a larger impact on asymmetry than training eccNet with MNIST), which seems to be less emphasized in the discussion than the role of the training set. Second, not all mechanisms are explored - some visual search effects are not reversed after training set rotation, suggesting some may arise from different visual features.

The fact that the 90-degree rotation of the training set leads to an effect polarity reversal does appear to point at statistical regularities in the training set, which is in line with previous research suggesting e.g. familiarity as a potential factor in visual search asymmetry. However, the authors also mention potential limitations of ImageNet. I am curious as to whether effects from the other experiments could be disrupted through training set alterations, and what explains experiment E (it'd be interesting to see what would happen with a different training set, e.g. Places)

Minor points:

In Figure 4 there seem to be no error bars in the human data plots? Figure 5 could also have error bars (e.g. SEM across experiments). The number of subjects who took part in the psychophysics experiments doesn’t seem to be mentioned even in the appendix.

---------------------------------------

After response:

Thank you for addressing my points so thoroughly. Great to see a direct fixation comparison, as well as the new training regimes - look forward to seeing the updated paper!

**Time Spent Reviewing:**

5

---

> ### Author Response · Authors · 2021-08-11
> **We thank the reviewer for useful feedback and interesting suggestions**
>
> 1.1. **[Fixation predictions]**
>
> This is an interesting question. We focused in the paper on the keypress responses because we did not have eye-tracking data for all the experiments.
>
> For the T versus L experiment, we include a comparison of fixations predicted by ECCNet versus human fixations below, as suggested. We compare the fixations in terms of two metrics: (i) number of fixations required to find the target in each trial
> (the cumulative probability distribution p(n) that
> the subject or model finds the target within n fixations);
> and (ii) scanpath similarity score, which compares the spatiotemporal sequence of fixations (Borji and Itti,  IEEE Trans PAMI 2013, Zhang et al, Nature Communications 2018). The results  show that the model approximates the fixations made by humans on a trial-by-trial basis  both in terms of the number or fixations and scanpath.
>
> Anonymous link for figures: [ResultsOnEyeMovementsForTvsL](https://drive.google.com/file/d/1XPVnmUI5Zb9ehL-6qdHETQCCSlRmUOTQ/view?usp=sharing)
>
> We will also include these results in the revision.
>
> The model and humans are similar but clearly not identical. These results are consistent with other  studies (which do not focus on search asymmetry) that provide a more extensive comparison of machine vs. human fixations during visual search (e.g., Zhang et al.  Nature Communications 2018).
>
> 1.2. **[Discussion emphasis]**
>
> We agree with the reviewer. The results suggest that *both* the model's architecture (multiscale top-down modulation, eccentricity-dependent sampling) and the training set are important to lead to asymmetry. We did *not* mean to imply that the only thing that matters is the training set.
> We will rewrite the discussion to note that both components are important.
>
> 1.3. **[Other aspects of training set statistics]**
>
> We thank the reviewer for these interesting suggestions. Because of time constraints during the rebuttal period, we could not examine the impact of the training set for all the experiments, but we did evaluate training using the Places dataset, as suggested for the reviewer. We will include this as well as other changes to the training set (see below) in the revised version.
>
> In FigS9, we reported the results in the six experiments after the model was trained using images that were rotated 90 degrees. As the reviewer correctly pointed out, 90-degree rotation is *not* the only explanation for asymmetry. Thus, only some of the asymmetry polarities in the six experiments were reversed.
>
> The point was a proof-of-principle demonstration of how a deep convolutional neural network computational model can reveal visual search asymmetry similar to humans and thus provide a plausible mechanism for asymmetric behavior. These results suggest that humans and machines may share some inherent biases due to architecture and development/training.
>
> There are a few interesting points indirectly alluded to in the reviewer's question. We would like to comment on them briefly here.(i) Is the mechanism for *all* forms of search asymmetry the same? We suspect that the answer is no. Thus, it is especially interesting to study the mechanisms behind other forms of asymmetry as the reviewer suggests.
> (ii) What constitutes adequate evidence in support of a model of search asymmetry? Given the assumption in (i), there may *not* be a single explanation or model for every form of asymmetry. While the accumulation of supporting evidence in the form of multiple different examples is instructive, the work here does *not* constitute a mathematical proof that the model can extrapolate to other forms of asymmetry, nor do we make such a claim. In addition to testing other forms of asymmetry we will clarify these points in the revised version.
>
> As the reviewer suggested, we tested EccNet on the Places dataset and rotated Places dataset.
>
> See anonymous link [RotatedPlacesAsFig4](https://drive.google.com/file/d/1a5AGIJVQZF1Ys1uJgrR4ewiRGMAczVw_/view?usp=sharing).
>
> We will add this figure in the revision.
> Consistent with the reviewer's prediction, this training altered some of the asymmetry polarities but not others.
> Compared with our observations in rotated ImageNet vs ImageNet shown in Experiment B lighting condition, similar reversal of asymmetry polarity was observed when training using the Places dataset. This implies that the feature statistics, commonly exists in all natural images, is one of the factors contributing to asymmetry.
>
> These results further support the idea that the training set has an important role in dictating asymmetry. The reviewer nicely connects this observation to familiarity, which is correct, see for example, Bruce and Tsotsos, 2011. We will add these results and discussion.
>
> The reviewer's question also inspired us to come up with different training regimes. One of them includes using a fisheye distortion (which reduces the proportion of straight lines and increases the proportion of curves). The other one is to introduce vertical and horizontal lines, thus increasing the proportion of straight lines. We conjecture that these changes to the training set will have opposite effects on asymmetry during visual search. Unfortunately, we have not been able to finish training on these datasets during the rebuttal period, but we will also include those results in the revised version.
>
> 1.4. **[Error bars and number of subjects]**
>
> We agree with the reviewer. We will add error bars to all the panels in Figure 5. The model results in Figure 4 already have error bars, which are rather small, and we will make sure to work on the aesthetic to make them clearer. For the human results in Figure 4, unfortunately, the original studies did not report error bars in all the cases. We will report the error bars and number of subjects for the human studies in those cases when they are available.

---

### Decision · Program_Chairs · 2021-09-28

**Decision:**

Accept (Poster)

**Comment:**

The paper investigates visual search asymmetry using a computational model, which integrate both bottom-up visual recognition with target-dependent top-down cues. By comparing the model against human behavior in visual tasks, the work revealed classical perceptual properties can emerge in neural network models pretrained on ImageNet, without the need for task-specific training. The findings are scientific and useful for providing computational evidence for human cognition. All the reviewers think the paper is well-written and provides a thorough view about human and machine behavior synergy in visual search. The reviewers give extensive, detailed reviews about the manuscript and made numerous points for clarification and improvements. The authors did a good job in the rebuttal that strengthens the reviewers confidence, which led to the increase of scores. These responses should be included in the revision. There are still a few points that deserve further clarification. Reviewer CMyL is not convinced about a few model choices as detailed in the post-rebuttal review. Both Reviewer F9R3 and CJM4 want further clarification about the motivation of the work. The authors should address these points in the revision.

**Consistency Experiment:**

NeurIPS has a long history of experimentation. In 2014, NeurIPS ran an experiment in which 10% of submissions were reviewed by two independent committees to quantify the randomness in the review process. This year, we repeated a variant of this experiment to see how the quality of the review process has changed over time.  This paper was part of the experiment and was therefore assigned to two committees (consisting of reviewers, an Area Chair, and a Senior Area Chair) that reached independent decisions.  If both committees made the same recommendation, this recommendation was followed. If a single committee recommended acceptance, the paper was accepted (with the exception of a few cases in which the other committee identified what we considered a fatal flaw, e.g., an error in a key result).

Both committees reached the same decision: **Accept (Poster)**

The other committee assigned to the paper recommended **Accept (Poster)**.  You can find the other set of reviews, along with any follow up discussion with the authors here:
https://openreview.net/forum?id=ar85GL0N11